# Learning by Self-Explaining

## Abstract

Artificial intelligence (AI) research has a long track record of drawing inspirations from findings from biology, in particular human intelligence. In contrast to current AI research that mainly treats explanations as a means for model inspection, a somewhat neglected finding from human psychology is the benefit of self-explaining in an agents' learning process. Motivated by this, we introduce a novel learning paradigm, termed **Learning by Self-Explaining** (LSX). The underlying idea is that a learning module (learner) performs a base task, e.g. image classification, and provides explanations to its decisions. An internal critic module next evaluates the quality of these explanations given the original task. Finally, the learner is refined with the critic's feedback and the loop is repeated as required. The intuition behind this is that an explanation is considered "good" if the critic can perform the same task given the respective explanation. Despite many implementation possibilities the structure of any LSX instantiation can be taxonomized based on four learning modules which we identify as: FIT, EXPLAIN, REFLECT and REVISE. In our work, we provide instantiations of LSX for two different learner models, illustrating different choices for the various LSX components. We evaluate these via various metrics on several popular benchmark datasets. Our results indicate improvements via Learning by Self-Explaining on several levels: in boosting the generalization abilities of AI models, *e.g.*, in small-data regimes, but also in mitigating the influence of confounding factors as well as leading to more task-specific and faithful model explanations. Overall, our results provide experimental evidence of the potential of self-explaining within the learning phase of an AI model.

## 1 Introduction

Self-reflection is considered an important building block of human intelligence and a crucial component in the learning process and knowledge development of humans (Gläser-Zikuda, 2012; Ellis et al., 2014). In fact, one aspect of self-reflection—self-explaining—has been identified in several psychological studies as greatly beneficial for the overall learning, problem-solving and comprehension abilities of human subjects (Chi, 2018; Chi et al., 1981; 1994; Chamberland & Mamede, 2015; Belobrovy, 2018; Larsen et al., 2013; Kwon & Jonassen, 2011; Bisra et al., 2018). Accordingly, self-explanations act as a means of making initially implicit knowledge explicit and thereby allow for iterative and critical *self-refinement*.

Indeed, recent works in AI research have picked up on the idea of self-refining, either directly inspired by findings from human studies (Madaan et al., 2023) or otherwise motivated *e.g.*, by the potential of pre-trained large language models (LLMs), *e.g.*, on the topics of self-debiasing (Schick et al., 2021) and self-instructing (Wang et al., 2023). Although these works are quite specific in their form of self-refinement (*cf.* Pan et al. (2023) for a recent survey) and far from the general idea of self-reflection from human psychology, they provide valuable first steps for more *reflective AI*. However, none of these focus on the value and potential of (self-)explanations as the basis and means of such reflective processes. On the other hand, research on interactive machine learning (Teso et al., 2023; Gao et al., 2022) such as explanatory interactive learning (XIL) (Teso & Kersting, 2019; Schramowski et al., 2020) has long identified the value of explanations as a means of communication between human users and AI models and particularly as a means for model refinement. However, hereby explanations are only leveraged for refinement through human guidance.

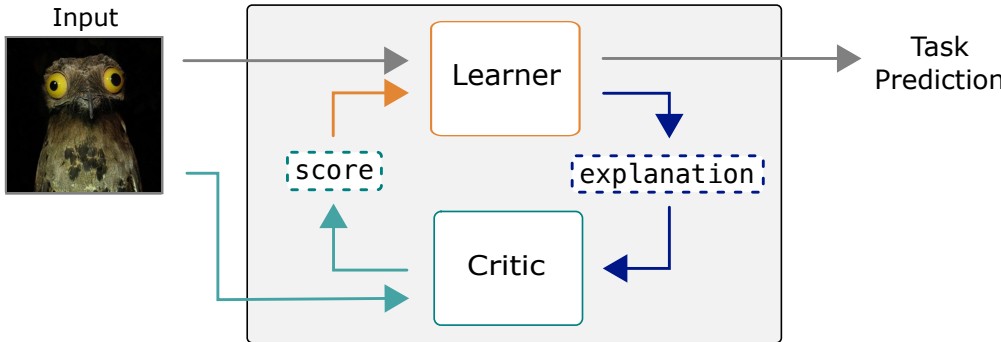

Figure 1: Learning by Self-Explaining (LSX) is a general learning approach, illustrated here in the context of image classification. It is characterized by two submodels, a *learner* and *critic*, and four distinct training modules: FIT, EXPLAIN, REFLECT and REVISE. Briefly, the learner is optimized for a base task in FIT, after which it provides explanations to its decisions in EXPLAIN. In the REFLECT module these explanations are passed to the critic, which "reflects" on the quality of these explanations. In other words, the critic evaluates how useful the explanations are for performing the base task. The resulting feedback from the critic, score, is used to update the learner's representations in REVISE. This loop can be repeated as needed.

In this work, we therefore introduce a novel machine learning paradigm called Learning by Self-Explaining (LSX) which leverages explanations in the learning phase of an AI model prior to any form of explanatory human guidance. The main idea is that an AI model consists of two submodels, a *learner* and a *critic*. The learning process in LSX is characterized by four learning modules sketched in Fig. 1. The learner is trained on a base task in FIT, after which it provides explanations for that task in EXPLAIN. The critic next performs the same task as the learner, but receives the input *and* corresponding explanations, thereby assessing the quality of these explanations for performing the original task. Intuitively, a "usefull" explanation should thereby provide important information for the task at hand. Finally, the critic's feedback is returned to the learner for revision in the REVISE module and the EXPLAIN, REFLECT and REVISE loop is repeated, if needed[1]. In the context of introducing LSX, we further present two instantiations of LSX for training a convolutional neural network (CNN) and the neuro-symbolic concept learner of Stammer et al. (2021) (NeSyCL), thereby illustrating specific configurations of the submodels and learning modules of LSX as well as the flexibility of the framework. In the context of our experimental evaluations on multiple benchmark datasets, we show that LSX boosts the generalisability of these base learners, particularly in the small data regime. Moreover, we show that LSX helps the base learner's of our instantiations mitigate the influence of confounding factors and leads to more consolidated, task-specific and faithful model explanations.

In summary, our contributions are the following: (i) We introduce a novel learning paradigm for machine learning, based on a model self-refining itself by evaluating its own explanations. (ii) We introduce two different instantiations of LSX, illustrating different submodel and learning module configuration choices. (iii) We provide extensive experimental evidence on various datasets and evaluation metrics, illustrating the ~~benefits and~~ potential of LSX.

We proceed as follows. In section 2, we formally introduce the LSX paradigm. In section 3, we next introduce two specific instantiations that integrate LSX into their training procedure. In our experimental evaluations in section 4, we provide results on various datasets and metrics illustrated via both LSX instantiations. We finish our work with an extensive discussion on related work and leave the reader with a final conclusion.[2]

## 2  Learning by Self-Explaining (LSX)

LSX is not explicitly bound to any one type of model implementation, data domain or base learning task. ~~but can be considered a general learning approach for any base learner.~~ In this section, we therefore give an

---

[1]Specifically, within the context of LSX we consider the two submodels to constitute a collective model, whereby both work jointly to perform the same base task. The overall model thus performs learning by explaining to parts of "itself".

[2]Code will be made available soon.

overview of the basic set of modules that characterize LSX before continuing with two specific instantiations in the next section. Let us first provide the background notations.

For simplicity, we introduce LSX here in the context of supervised image classification as base task. More formally, let $x_i \in X$ be an image, with the full dataset $X := [x_1, ..., x_N] \in \mathcal{R}^{N \times H \times W}$, and with corresponding class label to each $x_i$ defined as $y_i \in \{1, ...K\}$. Hereby, $N$ represents the number of data samples, $H$ and $W$ the image dimensions and $K$ the maximal number of image classes. Furthermore, let $X$ be split into a train and test split $X = \{(X_{train}, y_{train}), (X_{test}, y_{test})\}$ with $y_{train}$ representing the set of corresponding image class labels of the training set and $y_{test}$ those of the test set. The learner is provided the tuple set $\bar{X}_f = (X_f, y_f) = (X_{train}, y_{train})$ where we denote $y_f = y_{train}$ specifically as the label set provided to the learner. The critic set, $\bar{X}_c$, can represent a subset of $\bar{X}_f$ $e.g.$, $\bar{X}_c \subseteq \bar{X}_f$, but also a previously separated set from the test set (details below). Also for the critic we specifically denote $y_c$ as the label set of the $\bar{X}_c$.

Moreover, let us denote the two submodels: the learner as $f$ and critic as $c$. The learner can represent any desired learning model suited for the base task, e.g. a convolutional neural network (CNN). The critic, $c$, can in principle also be instantiated by any AI model, though this choice should be coordinated, among other things, with the choice of learner model (as will be discussed below). The general procedure of LSX can be described via four modules FIT, EXPLAIN, REFLECT and REVISE, where the last three modules describe the core of LSX and can be repeated until an iteration budget $T$ is reached. Let us now describe these four modules (presented in pseudo-code in Alg. 1 in the App.) in more detail.

**Base task** (FIT). The FIT module describes the underlying, base learning task in which the learner is optimized to solve a particular problem, $e.g.$, supervised image classification. Overall, this module is independent of LSX as it represents the standard, data-driven machine learning approach. More specifically, in FIT, $f$ is provided with the training dataset, $\bar{X}_f$, makes predictions given the base task, $\hat{y}_f = f(X_f)$, and optimizes its latent representations given the corresponding loss function, $l_B$. This loss function can $e.g.$, correspond to the cross-entropy loss, $l_B = l_{CE}(\hat{y}_f, y_f)$ when considering supervised classification. However, the underlying task can correspond to any form of learning ($e.g.$, other forms of supervision) and can contain any bells and whistles of modern machine learning setups ($e.g.$, hyperparameter optimization, learning rate schedulers, etc.). Finally, FIT returns a model optimized for the base task, $f = \text{FIT}(f, \bar{X}_f)$.

**Obtain explanations** (EXPLAIN). EXPLAIN represents the first of three core modules of LSX. In this module $f$ provides explanations to its decisions given a set of data samples, $X_c$. This is achieved via a predefined explanation method which returns an explanation for each sample, $E_c = \text{EXPLAIN}(f, X_c)$, where $E_c := \{e_1, ..., e_{|X_c|}\}$. If $\bar{X}_c$ contains ground-truth annotations, explanations can be queried given the ground-truth labels ($E_c = \text{EXPLAIN}(f, X_c|y_c)$), otherwise the learner provides explanations given its predictions. Hereby $\bar{X}_c$ can be a subset of $\bar{X}_f$ ($i.e.$, $\bar{X}_c \subseteq \bar{X}_f$), but can also represent a separate set that is with-held during the initial optimization of $f$ ($i.e.$, $\bar{X}_c \cap \bar{X}_f = \emptyset$). Overall, this explanation module can be realized with any explanation method from the vast literature of XAI ($cf.$ Guidotti et al. (2018); Ras et al. (2022b); Liao & Varshney (2021); Carvalho et al. (2019)). Given the architectural constraints of the learner, it can, $e.g.$, be an explanation in form of an importance map that indicates the degree by which a part of the input was important for a model's decision, an inherent concept-based explanation representing the activation of specific concepts that are present in an input, but also a generated natural language explanation, to name a few. Notably, this module is commonly also found in XIL (Friedrich et al., 2023a) approaches. In summary, this module returns explanations, $E_c$, from the learner.

**Reflect on explanations** (REFLECT). In the second core module, and arguably most distinctive module of LSX, the high-level role of the critic is to "reflect" on the quality of the explanations. This is an abstract measure and in LSX is quantified by the ability of the critic to perform the base task, given the learner's explanations. This explanation evaluation is performed in the REFLECT module whereby the critic returns a score of the learner's explanations, given $E_c$ and the corresponding original data that was used for generating these explanations, $\bar{X}_c$. In other words, score represents an indication of how well the critic performs the base task on the data $\bar{X}_c$ given the additional knowledge of the provided explanations, $E_c$ (an idea that is related to (Pruthi et al., 2022)). What score exactly represents depends very much on the model type of the critic. For example, it can represent the signal from a classification loss function over $\bar{X}_c$ and $E_c$ or a scoring of how probable an explanation is given the evidence of $\bar{X}_c$. By evaluating the quality of

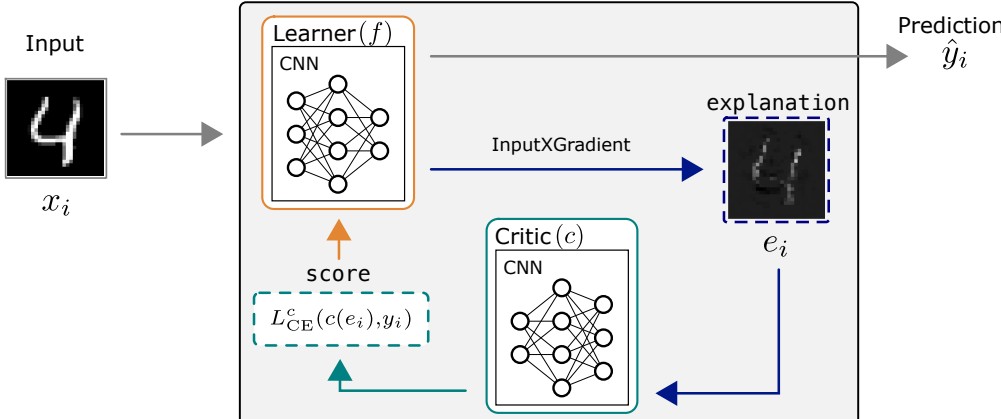

Figure 2: CNN-LSX: Learning by Self-Explaining instantiation for training CNNs for supervised image classification. Here CNNs represent both the *learner* and *critic*. Explanations are generated via InputXGradient. The `score` represents the classification loss of the critic on these explanations.

explanations based on their benefit in solving a task, the Reflect module represents one of the core aspects of LSX. Overall, explanations are thus treated not just as a one-way approach for indicating the importance of features when making a prediction (as often done in XAI). Rather, LSX specifically contributes to the view of explanations as verifiable rationales as in Fok & Weld (2023).

**Integrate feedback on explanations** (Revise). In the last module of an LSX loop, the feedback signal, `score`, obtained from the critic in the Reflect module, is used to refine the learner within Revise. This revision module can be realized in one of the many ways provided in interactive learning settings (Teso & Kersting, 2019; Friedrich et al., 2023a). Standard revision tools entail loss-based methods (Ross et al., 2017; Selvaraju et al., 2019) as well as more parameter-efficient finetuning alternatives (Houlsby et al., 2019b; Hu et al., 2022). But also non-differentiable revision approaches can be considered, *e.g.*, data augmentation approaches (Teso & Kersting, 2019) or retrieval-based setups that store the revisory feedback in an explicit revision engine (Friedrich et al., 2023b; Tandon et al., 2022). In our LSX instantiations, we implement the revision step by adding an additional explanation loss, $l_{\text{expl}}$ (*e.g.*, a HINT-like loss (Selvaraju et al., 2019) and a cross-entropy loss of the critic's classification) to the optimization process. The learner is thus jointly optimized in Revise via $L = l_{\text{B}} + \lambda l_{\text{expl}}$, where $\lambda \in \mathbb{R}$ represents a scaling factor.

## 3 Instantiating LSX

To give a better understanding of these abstract modules and their interdependencies, in this section, we introduce two instantiations of how to integrate different base setups into LSX based on the four modules (i - iv) described above. These instantiations will also be the point of investigation in our experimental evaluations.

### 3.1 Neural and Neurosymbolic Instantiations

**CNN-LSX via differentiable feedback score.** We begin with a CNN-based setup shown in Fig. 2 (*cf.* App. D.1 for formulas and further details). This instantiation which we denote as CNN-LSX consists of a standard CNN as learner, $f$, and a duplicate of the learner as critic, $c$. (i) Specifically, in this instantiation the learner is trained on raw images to predict the corresponding class labels and is optimized via a cross-entropy loss as $l_{\text{B}} := l^f_{\text{CE}}(f(X_f), y_f)$, thus representing the Fit module. (ii) The explain module is based on the post-hoc, differentiable InputXGradient method (Shrikumar et al., 2017; Hechtlinger, 2016) and we compute $E_c = \textsc{Explain}(X_c|y_c)$ with $\bar{X}_c \subseteq \bar{X}_f$. Specifically, an explanation via InputXGradient represents importance values per input pixel for a prediction (*cf.* the example in the dashed blue box in Fig. 2). Such explanations are also considered *local explanations* as they correspond to an explanation per individual

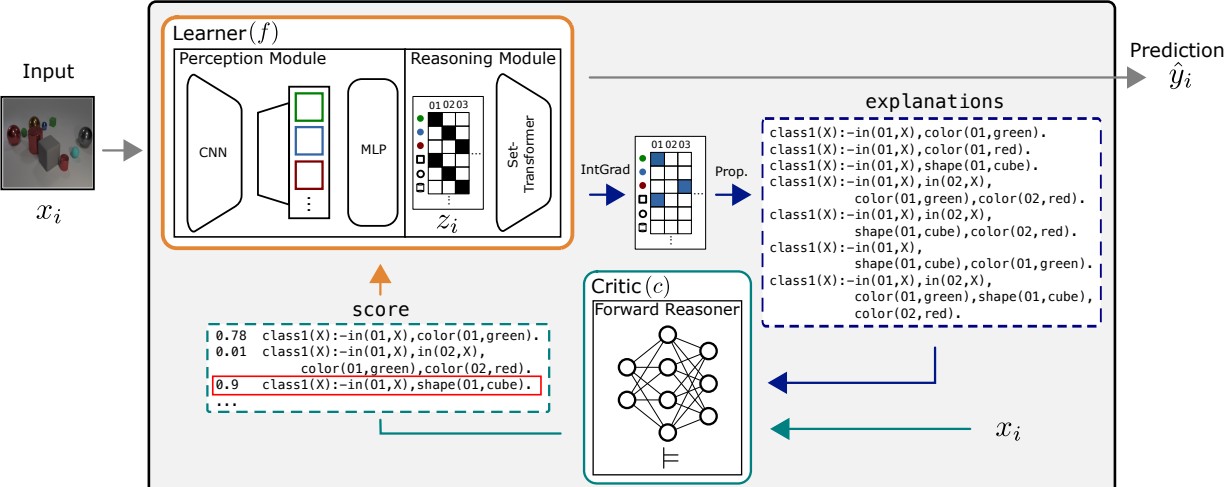

Figure 3: NeSyCL-LSX: Learning by Self-Explaining instantiation for supervised image classification via neuro-symbolic concept learner. The learner proposes a set of candidate class-specific logical explanations that is derived from concept-level importance maps. The critic represents a neuro-symbolic forward reasoner, which computes the validity of these logical statements given visual input. The feedback `score` represents a probabilistic ranking of the set of such logical explanations with which we identify the most likely explanation per image class and revise the learner to only use this explanation for samples of that class.

prediction. Specifically, InputXGradient is based on taking the partial derivatives of the output with respect to the input and multiplying this with the input.

(iii) In the REFLECT module, the critic next receives these explanations as input[3] and its role is to predict the corresponding class labels from these $\hat{y}_c = c(E_c)$ *i.e.*, can the critic predict the corresponding digit class "four" from the explanation in Fig. 2. The quality of these predictions is quantified via a second cross-entropy loss that is averaged over all samples in $E_c$. This is denoted as $l_{\text{CE}}^c(c(E_c), y_c)$ and represents the `score` within CNN-LSX. Intuitively, as the critic performs classification on the learner's local `explanations` the critic must identify common aspects across explanations that were provided for the same class. (iv) Next, in REVISE, $f$ is optimized via the joint loss $L = l_{\text{CE}}^f(f(X_f), y_f) + \lambda l_{\text{CE}}^c(c(E_c), y_c)$ for one epoch over $X_f$. In other words, the learner's parameters are updated based on both classification losses: the one from the learner given the training images $X_f$ *and* that from the critic given the explanations of $X_c$. Finally, we iterate over EXPLAIN, REFLECT and REVISE until an iteration budget, $T$, is reached.

**NeSyCL-LSX via non-differentiable feedback score.** Next, we will introduce an instantiation around the neuro-symbolic concept learner (NeSyCL) of Stammer et al. (2021) as base model (learner). We denote this instantiation as NeSyCL-LSX (*cf.* Fig. 3 and App. D.2 for formulas and further details). The NeSyCL consists of a slot attention-based perception module (Locatello et al., 2020) and set-transformer-based reasoning module (Lee et al., 2019). An image, $x_i$, is first processed by a pretrained perception module into a symbolic representation, $z_i \in [0, 1]^{O \times A}$ which indicates the presence of objects and their attributes in the corresponding image. Here, $O$ indicates the number of objects (or slots) and $A$ the number of predicted attributes. The reasoning module next makes a final task prediction given $z_i$. (i) As in the CNN-LSX instantiation, NeSyCL-LSX performs supervised image classification (base task) with $l_B$ representing a cross-entropy loss in the FIT module.

(ii) In comparison to CNN-LSX, however, where the learner provides local explanations, NeSyCL-LSX ultimately provides class-based explanations, *i.e.*, explanations to all predictions of a class. Specifically, `explanations` in NeSyCL-LSX represent class-specific *logical* explanations and obtaining these via the EXPLAIN module is a multi-step process. Initially, an importance map of $z_i$ is created via Integrated Gradients (Sundararajan et al., 2017) (a gradient-based method similar to InputXGradient above) for all

---

[3]Note, as an InputXGradient `explanation` also contains the input, it is not required to separately pass $X_c$ to the critic.

provided samples $x_i \in X_c$, (as in Stammer et al. (2021)). The resulting importance map, denoted here as $e_{z_i} \in [0, 1]^{O \times A}$, thereby indicates which objects and which of their attributes are relevant for an individual prediction. From these, we next transform the explanations into the form of logical statements. Specifically, we threshold the initially continuous-valued importance map (resulting in $e'_{z_i} \in \{0, 1\}^{O \times A}$) and for each subset of the indicated important attributes in $e'_{z_i}$ we formalize the set of all logical conjunctions[4]. We denote this overall set of extracted logical statements per sample as $e_i$. *E.g.*, as in Fig. 3, if $e'_{z_i}$ indicates that for sample $x_i$ it is important that one object is a *green cube* and one object is *red* the propositionalization step will extract the set of logical rules as shown in the dashed blue box in the figure (here in the notation of Shindo et al. (2023)). Next, we iterate over all samples of a class $k$ and group the collective $L_k$ logical explanations into a class set, denoted as $\hat{E}^k$. By iterating over all classes in this way we receive a set of sets of candidate explanations per class, $\hat{E}_c = \{\hat{E}^1, ..., \hat{E}^K\}$ which represents the $E_c$ of NeSyCL-LSX (in accordance with the notation of Sec. 2).

(iii) Next, the role of the critic in the REFLECT module is to rank each logical statement of $\hat{E}_c$ by its validity as an underlying class rule of the data in $X_c$. Specifically, the critic in NeSyCL-LSX is based on the neuro-symbolic forward reasoner of Shindo et al. (2021; 2023). Within NeSyCL-LSX this forward reasoner evaluates per class $k$ how likely a logical explanation within $\hat{E}^k$ represents a logical statement that is present in the images of $X_c$ that belong to class $k$. In other words, given the candidate rules of Fig. 3 the critic evaluates whether the attribute combinations of a specific statement are present among the objects in all $x_i \in X_c$ of the corresponding class. It then provides a probability of this explanation, where we denote $\rho^k \in [0, 1]^{L_k}$ for the set of these probabilities for all candidate explanations of $\hat{E}^k$. Finally, the set of these probabilities over the classes, $P = \{\rho^1, ..., \rho^K\}$, represents the `score` of NeSyCL-LSX.

(iv) Within the REVISE step the probabilities of the previous step, P, are first used to identify the most probable explanation per class $k$ (*cf.* red box in Fig. 3), denoted as $\hat{e}^k_{\max}$, and, consequently, $\hat{E}_{\max} = \{\hat{e}^1_{\max}, ..., \hat{e}^K_{\max}\}$ for the set over all classes. These most-probable explanations that are still in the form of logical statements are next transformed back into binary vector form in the dimensions of the learner's original symbolic representation denoted as $E'_{\max} = \{e'^1_{\max}, ..., e'^K_{\max}\}$ with $e'^k_{\max} \in \{0, 1\}^{O \times A}$. Finally, the learner, $f$, is enforced to provide this most likely explanation for each training sample of the corresponding class. As in Stammer et al. (2021) this is done via an additional mean-squared-error loss between the learner's initial explanations, $e_{z_i}$ (*i.e.*, the continuous-valued importance maps), and $E^k_{\max}$. We specifically denote the set of these importance map explanations over all samples in $X_f$ as $E_f$ and the final joint loss as $L = l^f_{\text{CE}}(f(X_f), y_f) + \lambda l_{\text{MSE}}(E'_{\max}, E_f)$. Finally, we iterate over EXPLAIN, REFLECT and REVISE until an iteration budget, $T$, is reached.

**Configuration choices.** As mentioned, many instantiations of LSX are possible, each with their own specific module and configuration choices. The instantiations introduced in this work, however, already sketch some interesting aspects and differences which we wish to highlight here. The most fundamental difference between CNN-LSX and NeSyCL-LSX lies within their REFLECT modules, specifically how a `score` of the learner's `explanations` are computed. In CNN-LSX the `score` represents a differentiable signal, where in NeSyCL-LSX this represents a probabilistic ranking of logical explanations. This second form of critiquing allows to weigh explanations and *e.g.*, identify the most "useful" explanation. The first form of critiquing, on the other hand, allows to perform loss-based explanation fine-tuning. Related to this and concerning the EXPLAIN modules, where in CNN-LSX *continuous* input-level `explanations` are being processed, the logical `explanations` in NeSyCL-LSX are *discrete*. As an effect of this, the form of revision in the REVISE module differs. In CNN-LSX we can simply pass the revisory signal from the critic to the learner via a backpropagated classification loss. In NeSyCL-LSX, however, we enforce the identified, most-likely explanation. Additionally, in CNN-LSX the critic represents a duplicate CNN of its learner. In NeSyCL-LSX, on the other hand, the critic represents a different model type altogether compared to its learner.

## 4 Experimental Evaluations

In the following evaluations, we investigate the benefits of Learning by Self-Explaining with the help of the two instantiations, CNN-LSX and NeSyCL-LSX. Specifically, we compare the performances of LSX to

---

[4]This step of changing the representation of relational data is commonly referred to as is propositionalization.

the standard training setup (*i.e.*, supervised learning). Over the course of our evaluations, we will investigate the potential benefits of LSX concerning test-set generalization, explanation consolidation, explanation faithfulness and shortcut learning mitigation.

### 4.1 Experimental Setup

**Data.** We provide experimental results on four different datasets. Particularly, we evaluate CNN-LSX on the MNIST (LeCun et al., 1989) and ChestMNIST (Yang et al., 2023; Wang et al., 2017) datasets and NeSyCL-LSX on the concept-based datasets CLEVR-Hans3 (Stammer et al., 2021) and a variant of Caltech-UCSD Birds-200-2011 dataset (Wah et al., 2011), CUB-10 (*cf.* App. E). Overall, the number of training images in MNIST corresponds to 60k, in ChestMNIST to 78k, in CLEVR-Hans3 to 9k and in CUB-10 to 300 images. Finally, for investigating the effect of confounding factors as a form of shortcut learning (Geirhos et al., 2020), we evaluate accuracies on CLEVR-Hans3 and DecoyMNIST (Ross et al., 2017), a confounded variant of the MNIST dataset. In the confounded settings of CLEVR-Hans3 specific object attribute combinations are fully correlated at training time, but uncorrelated at test time. *E.g.*, all large cubes are gray in the train set, but have any color in the test set. In Decoy-MNIST, on the other hand, confounders represent grayscale boxes in the corners of the images. Though the position in any of the four corners is random for an MNIST digit class, specific grayscale values are tied to specific classes at train time, but are random at test time. Importantly, due to the design of both datasets a high train set accuracy and high test set accuracy indicates the model is independent of the confounding factors, where a low test set accuracy, but high train accuracy indicates the model is strongly influenced by the confounding factors. We particularly refer to App. E for more details on these datasets.

**Metrics.** We provide evaluations of LSX based on five metrics where we briefly describe these here and refer to App. F for details. **(1)** The first metric is the standard *classification accuracy* on a held-out test set. **(2)** For investigating the revised explanations via LSX we provide the classification accuracy of a linear, ridge regression model. This model is optimized to classify a set of the learner's explanations (given their corresponding ground-truth class labels) and finally evaluated on a held-out set of explanations. **(3)** We further provide a cluster analysis based metric over all explanations, similar to the Dunn index (Dunn, 1973; 1974). This metric, which we denote as *Inter- vs. Intraclass Explanation Similarity* (IIES), quantifies how similar explanations are within one class, but dissimilar between classes (lower values indicate better separability). For investigating whether the learner in fact makes a decision based on the reported explanations, we analyse the faithfulness (Hooker et al., 2019; Chan et al., 2022) of the learner's explanations via two metrics as introduced by (DeYoung et al., 2020), namely **(4)** *sufficiency* and **(5)** *comprehensiveness*. Both metrics measure the impact on the model's performances of removing specific parts of the input based on the explanations. For comprehensiveness, parts of the input are removed which correspond to important features as identified by the explanation. For sufficiency, parts of the input are removed which correspond to unimportant features as identified by the explanation. For continuous input settings (MNIST and ChestMNIST) we modify the computation of these two metrics slightly by subtracting the impact when randomly chosen features are removed. This way we compensate for the potential influence of such out-of-distribution input. Notably, this can lead to negative values. In both formulations, however, higher comprehensiveness and lower sufficiency scores are better. Both metrics are not normalized and provide a relative comparison.

**Setup.** In all evaluations, we compare the performances of the LSX instantiations with the performances of the base learners that were trained for the same overall number of epochs, but only in the standard supervised manner. These are denoted as CNN and NeSyCL. We evaluate CNN-LSX on the MNIST and ChestMNIST datasets and NeSyCL-LSX on the CLEVR-Hans3 and CUB-10 datasets. The baselines were trained on the same data as the LSX versions, *i.e.*, $\bar{X}_f^{\text{baseline}} = \bar{X}_f^{\text{LSX}} \cup \bar{X}_c^{\text{LSX}}$. Note that for NeSyCL-LSX on CUB-10, we replaced the pretrained slot-attention module with a pretrained Inception-v3 network as perception module (Szegedy et al., 2016) and the reasoning module with a single linear layer as in (Koh et al., 2020). Hereby, the perception modules where pretrained to predict the symbolic representations, $z$, *i.e.*, relevant object attributes. These modules were used in the NeSyCL configurations both for the baseline as well as LSX configurations. Unless specifically noted, $\bar{X}_c$ represents a random subset of $\bar{X}_f$. We provide all results as mean values with standard deviations over five runs with different random seeds. Lastly, for investigating shortcut behavior we provide prediction accuracies on the unconfounded held-out test sets of

Table 1: Improved (few-shot) generalization via LSX on various datasets and models. We here present the accuracy in % on a held-out test set across varying training set sizes.

| MNIST | | | |
|---|---|---|---|
| | 1.2k | 3k | full (60k) |
| CNN | $89.83_{\pm 0.2}$ | $93.83_{\pm 0.08}$ | $\mathbf{98.70}_{\pm 0.1}$ |
| CNN-LSX | $\mathbf{91.59}_{\pm 0.91}$ | $\mathbf{94.31}_{\pm 0.43}$ | $98.03_{\pm 0.2}$ |
| ChestMNIST | | | |
| | 1.6k | 4k | full (78k) |
| CNN | $58.68_{\pm 0.15}$ | $58.49_{\pm 0.31}$ | $60.86_{\pm 0.08}$ |
| CNN-LSX | $\mathbf{61.16}_{\pm 0.54}$ | $\mathbf{61.77}_{\pm 0.75}$ | $\mathbf{63.41}_{\pm 1.3}$ |
| CLEVR-Hans3 | | | |
| | 180 | 450 | full (9k) |
| NeSyCL | $91.40_{\pm 1.80}$ | $96.81_{\pm 0.94}$ | $99.00_{\pm 0.28}$ |
| NeSyCL-LSX | $\mathbf{94.51}_{\pm 1.94}$ | $\mathbf{97.34}_{\pm 0.44}$ | $\mathbf{99.08}_{\pm 0.17}$ |
| CUB-10 | | | |
| | 100 | 150 | full (300) |
| NeSyCL | $83.57_{\pm 1.67}$ | $87.14_{\pm 0.4}$ | $93.13_{\pm 0.4}$ |
| NeSyCL-LSX | $\mathbf{84.49}_{\pm 1.18}$ | $\mathbf{93.05}_{\pm 1.72}$ | $\mathbf{96.33}_{\pm 0.31}$ |
| avg. improvement | **2.07** | **2.55** | **1.29** |

Table 2: Mitigating confounders via LSX: Test set performances on confounded datasets, both with deconfounded samples during training (*w/ deconf.*) and without (*w/ conf.*).

| DecoyMNIST | | |
|---|---|---|
| | w/ conf. | w/ deconf. |
| CNN | $63.52_{\pm 1.39}$ | $86.88_{\pm 0.68}$ |
| CNN-LSX | $\mathbf{78.99}_{\pm 2.71}$ | $\mathbf{88.43}_{\pm 2.34}$ |
| CLEVR-Hans3 | | |
| | w/ conf. | w/ deconf. |
| NeSyCL | $85.96_{\pm 4.20}$ | $91.23_{\pm 1.2}$ |
| NeSyCL-LSX | $\mathbf{90.90}_{\pm 4.38}$ | $\mathbf{95.64}_{\pm 2.21}$ |

Decoy-MNIST and CLEVR-Hans3 while being trained on the confounded train set. In the CLEVR-Hans3 evaluations that do not target evaluating the effect of confounding factors, the original confounded evaluation set of CLEVR-Hans3 was used as held-out test set. In the evaluations based on NeSyCL-LSX we set $T = 1$, where for those based on CNN-LSX $T > 1$. Lastly, we refer to App. D.1 and D.2 for details on the learner and critic data splits.

## 4.2 Experimental Results

**Improved (few-shot) generalisation.** An intuitive aspect of psychological findings on learning via self-explaining is that reflecting on one's learned explanations leads to improved generalizable knowledge (Chi et al., 1994). We investigate LSX in our first evaluation by measuring the held-out test set accuracy of CNN-LSX on the MNIST and ChestMNIST datasets and of NeSyCL-LSX on the CLEVR-Hans3 and CUB-10 datasets. In the rightmost column of Tab. 1, we present the respective test set accuracies for all learning configurations when trained on the full training size of each dataset. We can observe that on average, *i.e.*, over all datasets and over both LSX instantiations, there is a substantial boost in test set accuracy (last row). In the remaining columns of Tab. 1, we present the test-set accuracy in the smaller-data regime, *i.e.*, when the models were trained on different-sized subsets of the original training set [5]. We can observe large performance gains with LSX for all configurations and datasets. Particularly, these improvements are on average greater than those observed on the full training set sizes. Altogether these results suggest that learning via self-explaining can lead to improved test-set generalization performances, where we provide further analyses and a discussion on the potential limits of this effect in Sec. 5 and, importantly, App. C.

**Self-unconfounding.** In the second set of evaluations, we are interested in how far LSX can help mitigate shortcut behaviour (Geirhos et al., 2020). We particularly focus on confounded behaviour as a form of shortcut learning in which a learner picks up on spurious correlations within the training dataset that are not present in the test set (Schramowski et al., 2020). We investigate two settings. (i) With the first (denoted as *w/ conf.*), we investigate the performance of LSX in mitigating confounding behaviour without additional knowledge on the confounding factors. To this end, we train the two LSX instantiations (and baselines) as in the previous setups with $\bar{X}_c \subseteq \bar{X}_f$, and $\bar{X}_f$ representing a confounded dataset. (ii) In the second setting, $\bar{X}_c$ represents a dataset that is with-held from training the learner (*i.e.*, $\bar{X}_c \cap \bar{X}_f = \emptyset$) and represents a dataset that is explicitly deconfounded (*cf.* App. D.1 and D.2 for details on the sizes of $\bar{X}_c$). In other words, the spurious correlation found in $X_f$ is not present in $\bar{X}_c$. We denote this case as *w/ deconf.* For the standard training scheme the models have access to $\bar{X}_c$ within their training phase. We evaluate the CNN-

---

[5] Hereby, the size of these subsets varies over the different datasets due to different specifics of each dataset, e.g. the original training set sizes.

Table 3: Explanation consolidation via LSX. The metrics here are Inter- vs. Intraclass Explanation Similarity (IIES) of a learner's explanations (left) and the classification accuracy of a ridge regression model (RR. Acc., in %) on the learner's explanations (right). Both metrics are proxies for the explanation similarity within a class, yet separability between classes.

|  | IIES ($\downarrow$) | RR Acc. ($\uparrow$) |
|---|---|---|
| MNIST | | |
| CNN | $2.7_{\pm 0.07}$ | $12.32_{\pm 0.35}$ |
| CNN-LSX | $\mathbf{0.7}_{\pm 0.01}$ | $\mathbf{99.91}_{\pm 0.06}$ |
| ChestMNIST | | |
| CNN | $3.89_{\pm 0.13}$ | $74.87_{\pm 0.24}$ |
| CNN-LSX | $\mathbf{0.75}_{\pm 0.05}$ | $\mathbf{99.92}_{\pm 0.03}$ |
| CLEVR-Hans3 | | |
| NeSyCL | $0.65_{\pm 0.07}$ | $93.48_{\pm 2.41}$ |
| NeSyCL-LSX | $\mathbf{0.2}_{\pm 0.06}$ | $\mathbf{100}_{\pm 0.0}$ |
| CUB-10 | | |
| NeSyCL | $0.0266_{\pm 0.0005}$ | $100_{\pm 0.0}$ |
| NeSyCL-LSX | $\mathbf{0.0024}_{\pm 0.0001}$ | $100_{\pm 0.0}$ |

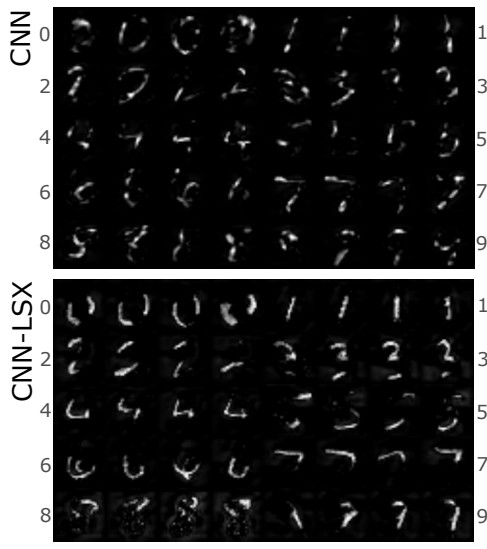

Figure 4: Exemplary explanations on MNIST of CNN baseline vs. CNN-LSX. Four random explanations are shown per image class (class ids on sides).

LSX configuration on the Decoy-MNIST dataset and the NeSyCL-LSX configuration on the CLEVR-Hans dataset.

In Tab. 2, we present the held-out test set accuracies of all configurations. We observe a strong improvement in performances when training LSX on the deconfounded critic sets (*w/ deconf.*), indicating that reflecting on the explanations of an explicitly deconfounded critic set can lead to much improved shortcut mitigation behaviour compared to the baseline learning setup.

Even more interesting is the result of the *w/ conf.* setting. We observe that the LSX trained models, though never having seen deconfounded data, lead to strong mitigation improvements. ~~This result suggests great practical implications, as it does not require prior knowledge on the confounding factors.~~ We hypothesize that the inductive biases of the specific explanation method and critic instantiation play a critical role for this effect. For example, the InputXGradient method (EXPLAIN) of CNN-LSX produces input-region-based explanations. The confounders in DecoyMNIST, on the other hand, represent gray-scale patches randomly placed in one of four corners where the digits themselves are rather consistently placed in the center of an image. Together with the results in Tab. 3 on more consistent, task-specific explanations this suggests that explanations based on more variable input-regions are more likely to be ignored via the reflection loop in LSX than explanations on the actual digits. We refer to our evaluations of Tab. 8 for additional evidence regarding this interpretation.

We additionally provide example explanations from both instantiations in App. F.3 which further support the findings of Tab. 2. Overall, our results suggest a beneficial effect of Learning by Self-Explaining in mitigating the issue of shortcut learning, specifically confounded behaviour.

**Explanation consolidation.** In the next evaluation, we wish to analyze how the critic's feedback signal influences the learner's representations, specifically its explanations. Based on the intuition behind the REFLECT module concerning "good" explanations, we hypothesize that the explanations of a LSX trained model represent more task-specific rationales. We present our results based on the Inter- vs. Intraclass Explanation Similarity (IIES) and the accuracy of a ridge regression model that was trained to classify a set of explanations and evaluated on a second, held-out set. Both of these metrics measure the class-based separability of a model's explanations. In Tab. 3, one can observe that over both metrics training via LSX leads to much more separable and distinct explanations. This effect appears less pronounced for the NeSy datasets, which is likely due to the sparseness and low dimensionality of their concept-level data and therefore

Table 4: Explanation faithfulness via LSX: Comprehensiveness and sufficiency results of explanations for models trained on all training samples.

| | Comp. (↑) | Suff. (↓) | | Comp. (↑) | Suff. (↓) |
|---|---|---|---|---|---|
| | MNIST | | | CLEVR-Hans3 | |
| CNN | $-1.34_{\pm 0.39}$ | $23.11_{\pm 1.18}$ | NeSyCL | $59.3_{\pm 3.6}$ | $21.67_{\pm 4.55}$ |
| CNN-LSX | $\mathbf{16.49}_{\pm 2.79}$ | $\mathbf{-0.21}_{\pm 4.18}$ | NeSyCL-LSX | $\mathbf{63.26}_{\pm 2.57}$ | $\mathbf{7.73}_{\pm 1.28}$ |
| | ChestMNIST | | | CUB-10 | |
| CNN | $13.98_{\pm 0.43}$ | $-4.2_{\pm 1.84}$ | NeSyCL | $64.54_{\pm 0.2}$ | $10.44_{\pm 0.45}$ |
| CNN-LSX | $\mathbf{18.84}_{\pm 0.38}$ | $\mathbf{-8.55}_{\pm 1.92}$ | NeSyCL-LSX | $\mathbf{64.59}_{\pm 0.23}$ | $\mathbf{6.3}_{\pm 0.34}$ |

also of the explanations. In Fig. 4, we also provide qualitative results of the explanation consolidation for MNIST, where explanations from four randomly sampled input samples are presented for each digit class. These visualizations undermine the quantitative results of Tab. 3 and particularly indicate the distinctness of the explanations from a LSX trained model within one data class from those of other classes. Overall, our results suggest that training via LSX can lead to more consistent explanations across samples of one class, yet distinctly separate explanations to samples of other classes. We refer to such an effect as *explanation consolidation.*

**Explanation faithfulness.** Although the performance improvements of the first evaluation suggest that LSX learners do make use of the critic's *explanatory feedback*, and the evaluations regarding explanation consolidation indicate that models learn more distinct explanations via LSX, an open question remains whether the learner's in fact make use of these *explanations* for making their decisions. In other words: are a learner's explanations faithful to its decision process? This is a relevant question on its own, particularly in the field of XAI (Hooker et al., 2019; DeYoung et al., 2020) and XIL (Schramowski et al., 2020) as models that produce unfaithful explanations are detrimental for building trust between human users and machines and at the same time make potential revisions via these explanations difficult (Schramowski et al., 2020; Teso et al., 2023). To investigate the faithfulness of LSX-learned explanations we turn to established faithfulness metrics of AI literature. Specifically, we use the *sufficiency* and *comprehensiveness* metrics of DeYoung et al. (2020). In Tab. 4, we present the results of these metrics over all four datasets and both LSX implementations[6]. One can observe a strong improvement via LSX in both metrics across all models and datasets. Specifically, the comprehensiveness results indicate that the information, considered as relevant by the explanations learned via LSX, are indeed important for the model to make its prediction. At the same time, the sufficiency results indicate that less important information based on the explanations has a decreased impact on the learner's decisions. Overall, these results suggest that training via LSX leads to more faithful explanations.

## 5 Discussion & Limitations.

Overall, our evaluations provide strong evidence for the benefits of training via LSX on a variety of important tasks and metrics that go beyond standard evaluations of ML research. This is further evidenced by initial results on LSX for visual-question answering based on large multi-modal models in App. B.

In the following, we wish to provide a general perspective for LSX on important current challenges of AI and conclude the section with an extensive discussion of the limitations of LSX.

**Human-machine interactions.** Accurate and trustworthy human-machine interactions have been identified as important criteria for the future deployability of AI systems (Friedrich et al., 2023a; Teso et al., 2023; Holzinger, 2021; Angerschmid et al., 2022). The LSX framework automatically facilitates the development and integration of mechanisms that allow for fruitful human-machine interactions. *E.g.*, via the EXPLAIN module a human user can query the learner's reasons for a prediction and via the REVISE module integrate feedback on these explanations into the model. Importantly, however, LSX does not remove the need for human-machine interactions for a sustainable model deployment. Specifically, there is no guarantee that a

---

[6]For CNN-LSX, we adapt these for handling continuous data (*cf.* App. F).

model that is trained via LSX is aligned with human requirements. Thus conclusive human assessment and potential revisions remain inevitable.

**System 1 and 2 processing.** A prominent hypothesis from cognitive psychology (which has gained recent interest in AI research (Goyal & Bengio, 2022; Kautz, 2022; Ganapini et al., 2022; Booch et al., 2021)) is that human cognition can be described via two processing systems: an approximate, fast system (system 1) that handles the majority of familiar situations and an embedded, slower, yet more exact system (system 2) that processes unfamiliar settings (Kahneman, 2011). There are interesting parallels between this framework and that of LSX where FIT can be considered to represent a fast, initial processing phase, and the triad consisting of EXPLAIN, REFLECT and REVISE to represent a slower, embedded processing phase. An important open question, particularly in AI research on system 1 and 2 processing, is on the form of communication between the two systems (Goyal & Bengio, 2022; Kautz, 2022). Explanations, as utilized in LSX, possess interesting properties for this aspect. Specifically, explaining and reflecting on the learner's explanations in LSX represents a form of making the *implicit* knowledge of the learner *explicit*. At the same time, system 2 processing can also influence the processing of system 1 as alluded to by our findings on explanation consolidation (*cf.* Tab. 3). Lastly, our NeSyCL-LSX has many parallels concerning the integration of neural and symbolic components to Henry Kautz's Neuro[Symbolic] system 1 and 2 approach (Kautz, 2022). Overall, however AI models are still far away from such models of human cognition and much additional research is needed.

**Limitations.** Despite the promising results of our instantiations there is still great potential for other design choices. Investigating such instantiations and their benefits is essential for consolidating the findings of this work. Specifically, providing a solid theoretical analysis will be of great value in the future. We here provide a discussion of several issues that can arise from insufficiencies in individual components of LSX and refer to detailed further evaluations in App. C.

Thus, although the results of Tab. 1 suggest benefits of LSX in terms of generalization, particularly in the context of small data, it is unlikely that this holds in all cases and instantiations. Particularly, if either the submodel's capacities are too low or the amount of data is too low (in contrast to the task and data complexity) it may not be possible for a learner to improve its overall predictive performance. Specifically, if a learner's predictive performance after the initial FIT phase lies around random guessing this can greatly influence the quality of the explanations: as the model has not identified important features within the data it will not provide distinct explanations to reflect these. In worst case, a resulting `explanation` can represent a distribution similar to uniform noise. Consequently, if the explanations are too bad and indistinguishable for the critic, also the critic can only perform at a level of random guessing such that the quality of the feedback to the learner will be of poor quality. We refer to Tab. 6 and Tab. 7 for detailed evaluations.

Furthermore, despite the interesting results on self-deconfounding and considering the additional evaluations of Tab. 8 in App. C it is important to conclude LSX as representing *one* potential tool in a toolbox of several mitigation strategies. Even more so, as combating the diversity and complexity of shortcut behaviour in general (Geirhos et al., 2020) does not allow for a one-size-fits-all approach. Lastly, the embedded processing, particularly of the REFLECT module, can lead to suboptimal computational factors for an LSX instantiation. Future work should investigate on more optimal forms of reflecting and revising than *e.g.*, introduced in CNN-LSX. Overall, the multitude of these analyses evidence that the notion of the no-free lunch theorem (Wolpert & Macready, 1997; Adam et al., 2019) also holds in the context of LSX and that it remains necessary for AI engineers to develop and assess the specifics of their LSX instantiations on a use-case basis.

# 6    Related Works

LSX is related to work in explainable AI (XAI), leveraging explanations in ML and, importantly, model refinement via self-refinement or feedback from a second model. Let us highlight these works in the following.

## 6.1    (Leveraging) Explanations in ML

Receiving explanations to an AI model's decision has become a heavily advocated and researched topic in recent years, culminating in the field of *explainable AI* (XAI) (*cf.* (Guidotti et al., 2019; Ras et al., 2022a; Roy

et al., 2022; Saeed & Omlin, 2023) for valuable overviews) and *interpretable AI*, which focuses on developing models that are *explicitly* interpretable by design (Räuker et al., 2023; Li et al., 2018; Rudin et al., 2022; Rudin, 2019). XAI methods, in general, are used to evaluate the reasons for a (black-box) model's decision by presenting the model's explanation in a hopefully human-understandable way. Current methods can be divided into various categories based on characteristics (Ras et al., 2022a), *e.g.*, their level of intrinsicality or if they are based on back-propagation computations. Across the spectrum of XAI approaches, from backpropagation-based (Sundararajan et al., 2017; Ancona et al., 2018), to model distillation (Ribeiro et al., 2016), or prototype-based (Li et al., 2018) methods, very often an explanation is created by highlighting or otherwise relating direct input elements to the model's prediction, thus visualizing an explanation at the level of the input space. Several studies have investigated methods that produce explanations other than these visual explanations, such as multi-modal explanations (Rajani et al., 2020), including visual and logic rule explanations (Aditya et al., 2018; Rabold et al., 2019). More recent work has also focused on creating concept-based explanations (Stammer et al., 2021; Zhou et al., 2018; Ghorbani et al., 2019).

An additional branch of research can be placed between explainable AI and interpretable AI, namely that of *self-explaining models* (Alvarez-Melis & Jaakkola, 2018; Lee et al., 2022; Roy et al., 2022; Camburu et al., 2018; Bastings et al., 2019; Majumder et al., 2022). In all of these works above and in contrast to LSX, explanations are only provided in a one-way communication as a means of model inspection for humans and not considered as a means of model refinement.

The idea of leveraging explanations in the training process has only recently been picked up by parts of the ML community. In the field of explanatory interactive learning (XIL) (Teso & Kersting, 2019; Schramowski et al., 2020; Stammer et al., 2021; Friedrich et al., 2023a) human users provide revisory feedback on the explanations of an ML model. Similar ideas can also be identified in other works of human-machine interactive learning (Teso et al., 2023; Gao et al., 2022), *e.g.*, in preference selection based interactions for learning vision language models (Brack et al., 2023). Compared to these, we argue for the importance of leveraging explanations in the training loop even before the necessity of human-machine interactions and advocate for the potential of explanations in a form of self-refinement in a model's initial learning process.

In contrast, several works have identified the value of leveraging explanations outside of human-interactive learning (*e.g.*, (Giunchiglia et al., 2022; Lampinen et al., 2021; 2022; Norelli et al., 2022)). In the works of Lei et al. (2016) and Bastings et al. (2019) (later categorized under the term *explain-then-predict models* by Zhang et al. (2021)), the goal is for one model to learn to extract the rationale[7] from an input and a second model to learn to predict the final class from these rationales. Similar ideas were picked up by (Zhang et al., 2021; Krishna et al., 2023). None of these works evaluate the correctness of explanations and particularly none use explanations as a means to *revise* a model.

## 6.2 (Self-)Refinement in ML

A recent, but quickly growing field of research related to our work is that which we categorize under the term of *self-refining AI*. This roughly encompasses research that investigates forms of self-supervised refinement of an AI model, *e.g.*, Wang et al. (2023) propose an approach for instruction-tuning. In the self-alignment approach of Sun et al. (2023), a LLM is aligned with few human provided principles. Schick et al. (2021), on the other hand, identify that LLMs can, to a certain degree, identify biases in their own generations and the authors leverage this characteristic in a finetuning process to mitigate biased generation in future prompts. In the work of Madaan et al. (2023) a model is used to provide feedback to its initial generations, where the feedback generation is guided via targeted, few-shot prompting. Zelikman et al. (2022), on the other hand, investigate finetuning a model by based on generated "chain-of-thought" rationales that lead to correct task predictions. Lastly, Paul et al. (2023) propose an approach in which a model learns to provide explicit intermediate reasoning steps for an answer via feedback from a critic model. Importantly in this work, the critic is specifically trained to identify false reasoning steps. In contrast to LSX only few of these mentioned approaches focus on refinement via explanations. Those that do require specifically trained modules for providing feedback on the explanations. In contrast in LSX explanations are quantified in how far they can

---

[7]Here we mean the term "rationale" as adopted in research on explainability in NLP.

help perform a task. Thus the evaluation and refinement of a model is performed without specific pretraining or prompt specification.

In contrast to self-refining AI a different branch of research focuses on revising a model based on forms of feedback from a second model. Such et al. (2020) which represents a meta-learning training data generation process in which a data generator and learner model are optimized for the same goal of improving the learner's performance on a given task. Nair et al. (2023) propose a general chat framework that leverages two agents, *researcher* and *decider*, to iteratively work through a task. The researcher plays the role of making task-specific suggestions to the decider, where the decider responds to the information provided by the researcher. In the student-teacher framework (Wang & Yoon, 2022) the goal is knowledge distillation, *i.e.*, learned knowledge from a trained model should be conveyed to a second model, the student model. Somewhat related to this is the concept of self-paced learning within the field of curriculum learning (Kumar et al., 2010; Wang et al., 2022) in which a model provides a signal on how "fast" to learn. Interestingly, Pruthi et al. (2022) frame the utility of an explanation in a student-teacher setup in which the goal is for a student model to simulate a teacher's behaviour best possible. Also Schneider & Vlachos (2023) argue for the importance of explanations in reflective processes. However, the authors only propose an approach where a model makes a final prediction based on the input and explanation that is estimated by a second model, similar to (Lei et al., 2016; Bastings et al., 2019; Zhang et al., 2021; Krishna et al., 2023). Overall, these approaches have a different target and motivation than our work. Particularly, in LSX the role of the critic submodel is to represent an internal optimization loop based on whether the explanations provided from the learner are beneficial in performing the original task.

## 7 Conclusion

In this work, we have introduced a novel learning framework, Learning by Self-Explaining (LSX), with which we argue for a novel perspective on the role of self-explaining in the process of learning in AI models. With this paradigm, we claim that explanations are important not just for human users to understand or to revise an AI model, but that they can play also an important role in a form of self-reflection in which an agent assesses its own learned knowledge via its explanations. Our experimental evaluations highlight several benefits of training via LSX in the context of generalization (across tasks and data modalities), knowledge consolidation, explanation faithfulness and shortcut mitigation. Conclusively, with this work, we provide evidence for the potential of explanations within a model's (self-)learning process and as an important step for developing more *reflective* AI.

**Future Research.** There are many avenues for future research related to LSX. *E.g.*, applying LSX to other modalities *e.g.*, natural language, where we have provided initial experimental results on visual-question answering via the vision-language model MAGMA (Eichenberg et al., 2022) in App. B. A more conceptual direction is the integration of a memory buffer of past LSX optimized explanations, allowing for models to re-iterate over previous generations of explanations (Chi et al., 1994). Additionally, integrating background knowledge into the explanation reflection process presents an interesting twist for LSX such that explanations are not just assessed based on the usefulness for the initial task, but also based on alignment with background knowledge. Another important view is the connection between self- and causal explanations (Carloni et al., 2023; Zečević et al., 2021; Heskes et al., 2020; Schwab & Karlen, 2019; Galhotra et al., 2021). Specifically, can an AI agent utilize its self-explanations to perform interventional and counterfactual experiments (Woodward, 2005; Beckers, 2022)? Another crucial avenue going forward is to further apply LSX to other forms of supervision, such as self-supervised learning or reinforcement learning approaches *e.g.*, integration into actor-critic approaches or for guiding curiosity driven replay (Kauvar et al., 2023).

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

# Appendix

In the following, one can find details on our LSX instantiations, experimental data and evaluation metrics and more.

## A    LSX in Pseudo-code

---

**Algorithm 1** Learning by Self-Explaining: Given two submodels, a learner model ($f$) and internal critic model ($c$), input sample sets $\bar{X}_f = (X_f, y_f)$ and $\bar{X}_c = (X_c, y_c)$ (tuple sets which include raw input samples *e.g.*, $X_f$, and corresponding label sets *e.g.*, $y_f$), original task (e.g. image classification) and iteration budget $T$.

---

1: $f \leftarrow \textsc{Fit}(f, \bar{X}_f)$ {Learner optimized for base task}
2: **repeat**
3:    $E_c \leftarrow \textsc{Explain}(f, \bar{X}_c)$ {Obtain explanations from learner for examples $X_c$}
4:    $\texttt{score} \leftarrow \textsc{Reflect}(c, E_c, \bar{X}_c)$ {Critic provides feedback on the quality of the explanations}
5:    $f \leftarrow \textsc{Revise}(f, \bar{X}_f, \bar{X}_c, \texttt{score})$ {Learner is updated given feedback from critic}
6: **until** budget $T$ is exhausted or $f$ converged
7: **return** $f$

---

## B    LSX for Visual-Question Answering

Table 5: Visual question answering accuracy (%) of a vision-language-based LSX instantiation (VLM-LSX) on the challenging VQA-X dataset. $\text{VLM}_0$ represents the pretrained VLM and VLM (ft.) the VLM finetuned on the VQA-X for question answering only.

| | $\text{VLM}_0$ | VLM (ft.) | VLM-LSX |
|---|---|---|---|
| VQA-Acc. (%) | 80.66 | 85.05 | **85.73** |

**Going beyond one modality.** In our last evaluations, we provide initial results of LSX applied to the challenging task of visual question answering (VQA) based on the vision-language model (VLM) MAGMA (Eichenberg et al., 2022). In addition to this *vision-language-based* learner, for this instantiation we employ a *language-based* critic model that thus provides feedback in only one of the learner's modalities. We refer to App. D.3 for further model and training details. In Tab. 5, we provide test-set question-answering accuracies on the VQA-X dataset (Park et al., 2018) for three different model configurations. $\text{VLM}_0$ represents the off-the-shelf MAGMA model (*i.e.*, zero-shot) and VLM (ft.) its VQA finetuned version. Lastly, VLM-LSX is finetuned on the VQA task *and* via explanatory feedback from the critic. One can observe that indeed LSX yields distinct improvements to MAGMA's VQA performance over the purely VQA-finetuned model, particularly when set in relation to MAGMA's off-the-shelf performance ($\text{VLM}_0$).

## C    Investigating Potential Failure Cases

In the following we wish to investigate potential failure cases of LSX. Particularly, although our main results indicate promising qualities of models that are trained via LSX insufficiencies in individual components can lead to poor overall model behaviour. We here provide a detailed discussion and evaluations of several issues where we focus on generalization in the context of small data and mitigating the effect of confounding factors.

In the following, we investigate three distinct cases that can influence the performance of models trained via LSX. Notably, these investigations are not all-inclusive, but provide an important assessment for future endeavours. The first case focuses on the issue of *too* small data. The second case focuses on the issue of the critic's capacity and its influence on the learner's performance. Lastly, we investigate the issue that can arise if the explanation method is inadequate in describing important discriminative features of the data.

Table 6: Additional small data evaluations with 300 samples for MNIST and 30 for CUB10.

| MNIST (300) | |
|---|---|
| CNN | $81.46_{\pm0.31}$ |
| CNN-LSX | $\mathbf{81.80}_{\pm0.13}$ |
| CUB-10 (30) | |
| NeSyCL | $\mathbf{66.02}_{\pm1.78}$ |
| NeSyCL-LSX | $63.82_{\pm3.96}$ |

Table 7: LSX via random critic feedback (w/ rand. $c$) on 3k MNIST and 150 CUB10 training samples.

| MNIST (3k) | |
|---|---|
| CNN | $93.83_{\pm0.08}$ |
| CNN-LSX | $\mathbf{94.31}_{\pm0.43}$ |
| CNN-LSX (w/ rand. $c$) | $93.06_{\pm0.14}$ |
| CUB-10 (150) | |
| NeSyCL | $87.14_{\pm0.4}$ |
| NeSyCL-LSX | $\mathbf{93.05}_{\pm1.72}$ |
| NeSyCL-LSX (w/ rand. $c$) | $87.24_{\pm2.93}$ |

Table 8: Investigating confounded behaviour on ColorMNIST.

| ColorMNIST | | |
|---|---|---|
| | w/ conf. | w/ deconf. |
| CNN | $\mathbf{59.34}_{\pm0.49}$ | $\mathbf{87.24}_{\pm1.52}$ |
| CNN-LSX | $57.62_{\pm0.49}$ | $46.01_{\pm3.21}$ |

**Too small data.**

With the first set of evaluations we wish to investigate the learner's performance in the case of more extreme data set sizes, particularly very small data sets. For this we provide further experiments with our two instantiations (CNN-LSX and NeSyCL-LSX) on only 300 MNIST samples (with 256 critic samples) for CNN-LSX and 30 CUB10 samples (with 30 critic samples) for NeSyCL-LSX.

The results are presented in Tab. 6 (test set prediction accuracies in %) and indeed suggest that the smaller the data set the less pronounced the effect becomes that we had originally observed in Tab. 1. Although we still observe improvements in the small data set regime with CNN-LSX, the results on CUB10 in Tab. 6 even appear to suggest that given a very small data set LSX can have a negative impact on the learner's predictive performance. This difference is most likely due to the discrete nature of explaining and revising in NeSyCL-LSX (*i.e.*, choosing the maximally probable explanation) such that if a bad explanation is chosen by the critic and enforced on the learner the learner's predictive performance will likely be negatively influenced by this. This is quite consistent with the findings of Friedrich et al. (2023a) on poor explanatory feedback in XIL. In comparison, the continuous nature of CNN-LSX seems to be more robust to this effect, though it is difficult to adequately compare the two data set sizes (300 MNIST vs 30 CUB10 samples) given the individual nature and complexity of the two data sets (grayscale images vs. concept-tagged images).

Interestingly, the point on "bad explanations" that cna potentially result from the critic also hints at a novel form of overfitting that is quite specific to LSX, as well as other forms of explanation guided learning: overfitted explanations (or other forms of explanatory feedback) can lead to poor predictive performances.

**Low capacity models.**

In the second set of evaluations we investigate the potential issues that arise from a critic submodel that performs poorly due to too low capacity. Specifically, we evaluate CNN-LSX and NeSyCL-LSX where we have replaced their original critic submodels with simulated critic that only performs random guessing on the base task (given the learner's explanations). We denote this setting as w/ rand. $c$. In Tab. 7 we present the test set prediction accuracies (in %) of these configurations on 3000 MNIST training samples for CNN-LSX and 150 CUB10 samples for NeSyCL-LSX where we have copied the original results of the baseline performances from the main text for easier comparison. We observe that indeed given a too low capacity of the critic the learner falls back to its base predictive performance for both instantiations. These results seem to suggest a lower bound for LSX due to random explanatory feedback.

Overall, it is conceivable that similar theoretical boundaries hold for LSX as they had been identified for boosting algorithms (Schapire, 1999; Freund & Schapire, 1997). Thus, if the learner can only perform prediction at the level of random guessing it is unlikely that the critic will be able to "use" the explanations. But if the learner performs even slightly better than random guessing and the explanations faithfully reflect such minute differences it is likely that the learner's performance can be improved given the right capacity of the critic and enough LSX iterations. However, a solid theoretical analysis of this will be required for conclusive remarks, where particularly the interplay between the different components in LSX will likely lead to interesting theoretical findings.

**Insufficient explanation method (for confounding factors).**

We lastly investigate the influence of the type of utilized explanations in LSX on a model's behaviour. Specifically, we adhere to investigating CNN-LSX's behaviour on a third confounded data set, ColorMNIST (Kim et al., 2019; Rieger et al., 2020). Briefly, this data set represents another confounded version of the benchmark MNIST digits. In contrast to DecoyMNIST, however, the confounding factor is not spatially separate from the relevant features. In ColorMNIST each digit of a specific class is correlated with a specific color within the training set, but uncorrelated at test time (*cf.* Fig. 2 of Stammer et al. (2021) for a visual overview). *E.g.*, all nines in the training set will posses a purple color but random colors at test time. Thus, if a model has learned to focus only on the color of the digits it will incorrectly predict the class of a *purple* zero at test time to be the class "nine".

As discussed in our confounding mitigation evaluations of Tab. 2 the inductive bias of the explanation method likely plays an important role in our observed mitigation effect of LSX. In this set of evaluations we therefore investigate the influence of an "insufficient" `explanation` method. We consider an explanation method insufficient in the context of confounders if an explanation from a model that focuses on the relevant features (*e.g.*, the digit shape) is not distinguishable from an explanation from a model that focuses on the confounding features (*i.e.*, the digit color in ColorMNIST). Specifically, the explanation method utilized in CNN-LSX, InputXGradient, represents an importance map that indicates which pixel of the input image is important for a prediction. However, a digit pixel could be important due to its color or position (digit shape). In other words such a type of explanation is too coarse to distinguish relevant from confounding features in the case of ColorMNIST and InputXGradient therefore represents an insufficient explanation method in this context.

Tab. 8 presents the prediction accuracies on the non-confounded test set both of the vanilla trained CNN and the LSX trained CNN for the two configurations w/ conf. and w/ deconf. (as in the previous confounding evaluations). We observe quite different results from those that were based on DecoyMNIST where we observe confounding mitigation behaviour neither for w/ conf. nor for w/ deconf.. To the contrary, we observe minor drops in accuracy for w/ conf. and even larger drops in accuracy particularly in the w/ deconf. condition. This suggests that when the explanation method does not adequately depict the level of detail that is required for the data and task, learning via such insufficient explanations can lead to a disadvantage for the overall model performance. Particularly, the explanatory feedback based on $X_c$ can lead to a negative impact on the learner's predictive performance when the learner's predictions and explanations are evaluated on differently distributed data splits as in the w/ deconf. case (*i.e.*, confounded $X_f$ for the task predictions vs non-confounded $X_c$ for the explanations), but the explanation method can not express these differences. However, modifying the explanation method *e.g.*, from an input-based to concept-based explanation method, will likely lead to beneficial mitigation effects as evidenced by NeSyCL-LSX (which utilizes a concept-based explanation method) on the concept-confounded CLEVR-Hans3 dataset in Tab. 2.

# D    Model and Training Details

Exact implementation details can be found in the corresponding code[8].

## D.1    CNN-LSX

The CNN-LSX configurations are identical for the MNIST and ChestMNIST dataset.

**Learner.**

The learner corresponds to a convolutional neural network (CNN) with two convolutional layers, ReLU activation layers, one average pooling layer and two linear layers.

**Critic.**

The critic for CNN-LSX is identical to the architecture of the corresponding learner for both MNIST and ChestMNIST.

---

[8]Complete code will be made available upon acceptance

Fit.

Within the FIT module the learner is optimized via a cross-entropy loss as $l_B := l^f_{CE}(f(X_f), y_f)$.

EXPLAIN.

The explanation method of CNN-LSX corresponds to the InputXGradient method described in Shrikumar et al. (2017) and implemented via the captum[9] pytorch package. Following Ancona et al. (2018), for an input sample $x_i$ and the output of model $f$ given the corresponding ground truth label, $y_i$, it is defined as:

$$e_i = x_i \cdot \frac{\partial f_{y_i}(x_i)}{\partial x_i}.$$

REFLECT.

In the REFLECT module for CNN-LSX the critic trains for one epoch on the explanations obtained from the learner given the input data of $\bar{X}_c$, *i.e.*, $E_c$. Specifically, the critic here is trained via a cross-entropy loss to predict the corresponding class of the learner's explanations. We allow the critic to update its parameters while iterating over all batches in $(E_c, y_c)$ whereby the loss values are accumulated over all batches and averaged. In practice we found that it was beneficial to reinitialize the critic with each LSX iteration. The final accumulated and averaged loss value is passed back to the learner and represents the feedback `score` in CNN-LSX.

REVISE.

In a standard REVISE step the learner again performs the original base task via $l_B$ while jointly optimizing for the feedback via the critic in the previous REFLECT step. Specifically, the learner optimizes a joint loss: $L = l_B + \lambda l^c_{CE}(c(E_c), y_c)$. Hereby, $\lambda$ represents a scaling hyperparameter which we set quite high (*e.g.*, $\lambda \geq 100$) in our evaluations to prevent the learner from mainly optimizing for good prediction. Also here we refer to the corresponding code for the exact parameter values.

We perform the triad of LSX modules (EXPLAIN, REFLECT, REVISE) for several iterations until iteration budget $T$ is reached.

As a final REVISE step, *i.e.*, when the iteration budget has been reached, we perform a fine-tuning step in which we let the learner produce explanations for all samples in $X_f$, $E_f = \text{EXPLAIN}(f, X_f | y_f)$, and let $f$ be optimized for the base task making sure that it does not diverge its explanations from $E_f$ in the process. This is done via the combined loss $L = l_B + \lambda_{ft} l_{ft}(E'_f, E_f)$, where $l_{ft}$ represents a simple mean-squared error loss between $E_f$ and the explanations that are generated within each optimisation iteration.

**Dataset ratios.**

For the results in Tab. 1 via CNN-LSX $\bar{X}_c$ presented about $\frac{1}{2}$, $\frac{2}{3}$ and $\frac{1}{2}$ of $\bar{X}_f$, from left column to right column, respectively.

For the results in Tab. 2 via CNN-LSX on Decoy-MNIST we present the critic with 512 samples from approximately 60000 training samples for *w/ conf.* and 1024 test set samples for *w/ deconf.*.

**D.2  NeSyCL-LSX**

**Learner.**

The learner submodel for the NeSyCL-LSX instantiation differs for the CLEVR-Hans3 and CUB-10 datasets. For CLEVR-Hans3 the learner corresponds to the concept learner of Stammer et al. (2021) which incorporates a slot attention encoder for predicting the object's attributes and a set transformer for the final class prediction. As in Stammer et al. (2021), in our experimental evaluations, we make use of a pretrained perception (slot attention) module and perform updates only to the reasoning (set transformer) module, *i.e.*, the module making the final predictions. For the CUB-10 configuration the learner corresponds to the

---

[9]captum.ai/

setup of Koh et al. (2020) representing an Inception-v3 model Szegedy et al. (2016) for predicting the bird concepts and a simple linear layer to make the final class prediction.

**Critic.**

The critic, $c$, of the NeSyCL-LSX instantiation, both for CLEVR-Hans3 and CUB-10, corresponds to the neural-symbolic forward reasoner of Shindo et al. (2021). Where the predicate specifications etc. required for the forward reasoner for CLEVR-Hans3 correspond to the original ones of Shindo et al. (2021). For CUB-10 we had to redefine each of the 28 bird concepts as neural predicates *e.g.*, `haswingcolor` can take six different values which in the notation of Shindo et al. (2021; 2023) is defined as: `haswingcolor:brown,grey,yellow,black,white,buff`. We refer here to our repository and the original work of Shindo et al. for details.

FIT.

Similar to the CNN-LSX instantiation the FIT module in NeSyCL-LSX corresponds to optimizing for class prediction via a cross-entropy loss $l_{\mathrm{B}} = l_{\mathrm{CE}}(f(X_f), y_f)$. As previously mentioned, in our evaluations we hereby freeze the parameters of the perception module of $f$, thus optimizing only the parameters of the reasoning (aka predictor) module of the learner.

EXPLAIN.

The EXPLAIN module of NeSyCL-LSX builds on the explanation approach of the concept learner of Stammer et al. (2021). Specifically, it first computes the integrated gradients (Sundararajan et al., 2017) of the symbolic representation, $z_i$, given the prediction. Following the notation of Ancona et al. (2018) for an input $z_i$ this is defined as:

$$\mathrm{IntGrad}_i = (z_i - \bar{z}_i) \cdot \int_{\alpha=0}^{1} \frac{\partial f_{y_i}(\tilde{z}_i)}{\partial \tilde{z}_i}\Big|_{\tilde{z}_i = \bar{z}_i + \alpha(z_i - \bar{z})} d\alpha.$$

Hereby, $\bar{z}_i$ represents a "baseline" value, which in our evaluations corresponds to a zero vector. Next, the resulting importance map on the latent concept representations, $e_{z_i} \in [0,1]^{O \times A}$, is binarized via a hyperparameter $\delta \in [0,1]$ to $e'_{z_i} \in \{0,1\}^{O \times A}$. We next propositionalize [10] $e'_{z_i}$ by representing the explanation as a set of logical statements that are present in $e'_{z_i}$. These logical statements consist of all subsets of conjunctive combinations of the important attributes and objects. We denote the set of these candidate logical explanations generated from sample $x_i$ as $\hat{E}_i$. For example, let us assume that for a specific CLEVR-Hans3 sample $x_i$ of class 1 we identify two objects to be important for the final class prediction. Hereby, the attributes *green color* and *cubical shape* are important for the first object and *red color* for the second object. Following the notation of Shindo et al. (2023) the set of generated candidate are:

```
class1(X):- in(O1,X),color(O1,green).
class1(X):- in(O1,X),shape(O1,cube).
class1(X):- in(O1,X),color(O1,red).
class1(X):- in(O1,X),color(O1,green),shape(O1,cube).
class1(X):- in(O1,X),in(O2,X),color(O1,red),shape(O2,cube).
class1(X):- in(O1,X),in(O2,X),color(O1,green),color(O2,red).
class1(X):- in(O1,X),in(O2,X),color(O1,red),color(O2,green),shape(O2,cube).
```

We refer to this step of constructing all potential candidate rules as propositionalizing.

Notably, each input sample thereby produces a set of such candidate rules which may potentially contain many duplicates over samples of the same underlying class. Finally, by iterating over all samples in $X_c$,

---

[10]Changing the representation of relational data.

grouping the resulting candidate rules by image class and removing duplicates we receive a set of candidate rules per class as $\hat{E}_c = \{\hat{E}^1, ..., \hat{E}^K\}$, where $\hat{E}^k$ denotes the set of generated candidate logical explanations gathered over all samples of class $k$ and with duplicates removed. This $\hat{E}_c$ represents the $E_c$ of the notation of Sec. 2.

For improved running times it is beneficial to limit the number of candidate rules per input sample by a maximum number of objects and attributes per object within an explanation rule $e.g.$, maximally four objects per rule. In our evaluations we set these two hyperparameters to still greatly overestimate the ground-truth rule and refer to the code for details (as well as for the values of $\delta$). It is important to note that the propositionalizing step breaks the differentiality of the explanations.

### REFLECT.

Having obtained the set of candidate `explanation` rules per image class, we pass these candidate rules to the critic $i.e.$, forward reasoner of Shindo et al. (2021). For each underlying class and based on the data within $X_c$ the critic next estimates the validity of each candidate rule, where we refer to Shindo et al. (2021) and Shindo et al. (2023) for details on this.

This evaluation is done for all positive examples of a class and for all negative examples of a class ($i.e.$, all remaining classes), resulting in two probabilities for the ith explanation candidate from set $\hat{E}^k$ of class $k$, which contains $L_k$ candidates in total. We denote these probabilities as $\rho^{k+} \in [0, 1]^{L_k}$ and $\rho^{k-} \in [0, 1]^{L_k}$, respectively. The first probability represents the validity of the rule as observed in samples only of the relevant class $k$ and the second represents the validity in samples of all other (irrelevant) classes ($j \in \{1, ..., K\} \backslash k$).

As we consider an explanation to be good if it distinguishes an input sample from samples of opposite classes, but indicates similarities to samples of the same class, we next compute the probability for each candidate logical explanation as $\rho^k = \rho^{k+} - \rho^{k-}$. The set of these probabilities over classes $P = \{\rho^1, ..., \rho^K\}$ represents the `score` of NeSyCL-LSX and represents the numerical values in the `score` representation in Fig. 3.

### REVISE.

Finally, per image class, $k$, we select the explanation rule with the maximal probability from $\rho^k$ corresponding to the red enclosed rule in Fig. 3. We denote this as $\hat{e}^k_{\max}$ with $\hat{E}_{\max} = \{\hat{e}^1_{\max}, ..., \hat{e}^K_{\max}\}$ for the set over all classes.

The selected logical explanations, $\hat{E}_{\max}$, are next mapped back into binary matrix form in the dimensions of the learner's latent symbolic representation $E'_{\max} = \{e'^1_{\max}, ..., e'^K_{\max}\}$ with $e'^j_{\max} \in \{0, 1\}^{O \times A}$. This is required so we can compare, in a differentiable manner, the learner's explanations to the valid explanations as identified by the critic. Specifically, we compare the explanations in $E'_{\max}$ with $E_f$ which represents the set of continuous-valued explanations at the level of the learner's symbolic representation, $e_{z_i}$ for $x_i \in X_f$. Thus, in the REVISE step of NeSyCL-LSX we optimize a joint loss function corresponding to $L = l_B + \lambda l_{\mathrm{MSE}}(E'_{\max}, E_f)$. For explanation $e_{z_i}$ of input sample $x_i \in X_f$ with corresponding class label $y_i$ $l_{\mathrm{MSE}}$ is defined as:

$$l_{\mathrm{MSE}}(e'^{y_i}_{\max}, e_{z_i}) = \frac{1}{O \times A} \sum_{q=1}^{O \times A} (e_{z_{i_q}} - e'^{y_i}_{\max_q})^2.$$

In comparison to CNN-LSX in our evaluations we set $T = 1$ for NeSyCL-LSX. This means the critic only scores the proposed underlying logical explanations once and passes this back as feedback. Although it is in principle possible to perform multiple steps of this, $e.g.$, by first removing explanations which are most *unprobable* from the learner's representations and only after several of such iterations choose the most likely explanation, we leave this for future investigations.

**Dataset ratios.**

In the setting of NeSyCL-LSX on CLEVR-Hans3 for the results in Tab. 1 $\bar{X}_c \cap \bar{X}_f = \emptyset$. Specifically, from left column to right column the ratio between $\bar{X}_c$ and $\bar{X}_f$ was 30 to 150, 75 to 375 and 1500 to 7500, respectively. For the evaluations via NeSyCL-LSX on CLEVR-Hans3 the critic was provided 1500 samples from the original training set of 9000 samples.

The setting of NeSyCL-LSX on CUB-10 for the results in Tab. 1 represented a small variation from the previous settings in that $\bar{X}_f \subseteq \bar{X}_c$, due to the small number of samples per class in CUB-10 in combination with the neuro-symbolic forward reasoner (*i.e.*, the critic). In this way for the results in Tab. 1 for CUB-10 from left column to right column the ration between $\bar{X}_c$ and $\bar{X}_f$ was 29 to 24, 149 to 124 and 300 to 249, respectively.

For the results in Tab. 2 via NeSyCL-LSX on CLEVR-Hans3 we present the critic with 50 samples and the learner 7500 separate training samples for *w/ conf.* and 150 test set samples for *w/ deconf.*

### D.3  VLM-LSX

**Learner.**

The learner submodel of the VLM-LSX instantiation corresponds to the pretrained vision-language model MAGMA (Eichenberg et al., 2022) which consists of a CLIP-based image encoder and GPT-based language model (LM).

**Critic.**

The critic represents a simulated language model (LM) that provides preference scores to a set of proposed textual explanations given the corresponding original input and its previously acquired knowledge. Specifically, as suggested in Brack et al. (2023) we utilize the set of annotated explanations of the original VQA-X dataset as a form of knowledge base such that the critic scores generated explanations based on this knowledge base (further details in REFLECT below).

**Fit.**

The FIT module in the VLM-LSX instantiation represents the pretraining of the original MAGMA model. Specifically, Eichenberg et al. (2022) trained MAGMA on a collection of large scale datasets for image-caption prediction. Overall, with VLM-LSX we consider image-question-answer tuples $(i, q, a)$ consisting of an image, $i$, and a respective pair of text sequences for the question, $q$, and answer, $a$. We thus cast the image-captioning task of MAGMA's pretraining in the FIT moduel in the same setup, with $a$ representing the annotated image caption and $q$ representing the question prompt for providing a caption. Let us denote $X_f = (I_f, Q_f)$ and $\bar{X}_f = (X_f, A_f)$ as the corresponding learner sets, where in our evaluations of VLM-LSX $\bar{X}_f = \bar{X}_c$. The base task of VLM-LSX is performed via a language modeling loss for next token prediction of the answer. This is based on the next-token log-probability and conditioned on the previous sequence elements (we refer to Eichenberg et al. (2022) for details). This next-token loss represents $l_B = l_{vqa}(f(X_f), A_f)$ and optimization is performed via adapter-based finetuning (Houlsby et al., 2019a).

**Explain.**

`Explanations` in VLM-LSX represent explicitly generated textual sequences. Specifically, we let the learner generate a set of $N_e \in \mathbb{N}$ explanations per sample, denoted as $E_i$ for sample tuple $(x_i, q_i, a_i) \in \bar{X}_c$. Here we follow the approach of Brack et al. (2023) for the explanation sampling process. Briefly this is based on top-k sampling and explanation prompt engineering, where an explanation prompt is the sequence of tokens appended to the image, question, and answer to elicit textual explanations. This process overall results in $N_e$ different natural language explanations per data sample.

**Reflect.**

Within the REFLECT model of VLM-LSX (and similar to NeSyCL-LSX) the critic provides preference `scores`, $\rho_i \in [0, 1]^{N_e}$, over the generated explanations. These scores are based on the explanation's usefulness for solving the base task. This preference scoring is based on calculating the sample-wise ROUGE-L score (Lin, 2004) between the learner's generated explanations and the annotated explanations that are stored in the critic's knowledge base. We further follow Brack et al. (2023) in setting the threshold of ROUGE-L $\geq 0.7$ as indicating a "good" explanation.

**Revise.**

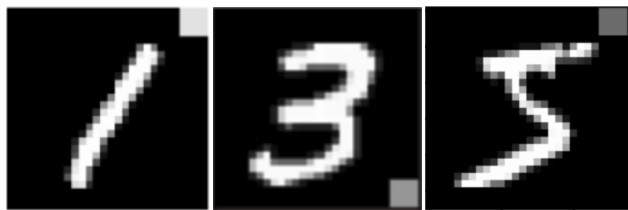

Figure 5: Example training images from Decoy-MNIST.

Based on the **score** of the previous Reflect step we next select the **explanation** from $E_i$ with the highest score. We denote $E_{\max}$ as the set of these maximally scored explanations over all samples within $\bar{X}_c$. We next add an additional loss to the learner's base loss: $L = l_B + l_{\mathrm{expl}}(f((I_c, Q_c, A_c)), E_{\max})$ and optimize the learner via adapter-based finetuning (as in the Fit module).

**Training details.**

In our training setup we set $N_e = 5$ and notably $\bar{X}_c = \bar{X}_f$. The number of LSX iterations was set to $T = 8$, where VLM (ft.) (*cf.* Tab. 5) was trained for the same number of overall steps.

## E  Data

**CUB-10.** CUB-10 represents a subset of the original Caltech-UCSD Birds-200-2011 dataset (CUB-200-2011) (Wah et al., 2011) that comprises images of the first 10 classes of the full dataset. Koh et al. (2020) originally perform a preprocessing step for CUB-200-2011 where concept vectors are replaced with the max voting vector over all samples of a class. In other words, the resulting concept activations are identical across all samples of a class which leads to a one-to-one mapping between concept activations and the class affiliation.

In CUB-10 we simulate a more realistic setting in which the class concept activations of (Koh et al., 2020) are overlaid with additional random noise, thereby maintaining the underlying class-based concept activation, but producing random variations per class sample. Specifically, we add uniformly distributed noise between 0 and 1 onto the class-based concept activations and binarize the resulting activations with a threshold of 0.75.

**DecoyMNIST**

DecoyMNIST (Ross et al., 2017) represents a version of MNIST (LeCun et al., 1989) in which small boxes are placed randomly in one of the four corners for each sample (*cf.* Fig. 5). Importantly, the dataset contains confounders. Within the training set the gray boxes possess a specific grayscale value for each digit class, where this grayscale value is randomized at test time.

**CLEVR-Hans3**

Fig. 6 presents the data distribution in CLEVR-Hans3 (Stammer et al., 2021). Specifically, CLEVR-Hans3 is based on the graphical environment of the original CLEVR dataset (Johnson et al., 2017), however reframed for image classification rather than visual question answering. Each class is represented by an underlying logical rule consisting of the presence of specific object combinations. Within the original training and validation set specific combinations of object properties are highly correlated, where they are not within the test set. *E.g.*, for the first class all large cubes are gray within the training set, but any color in the test set. This gray color thus represents a confounding factor within CLEVR-Hans3.

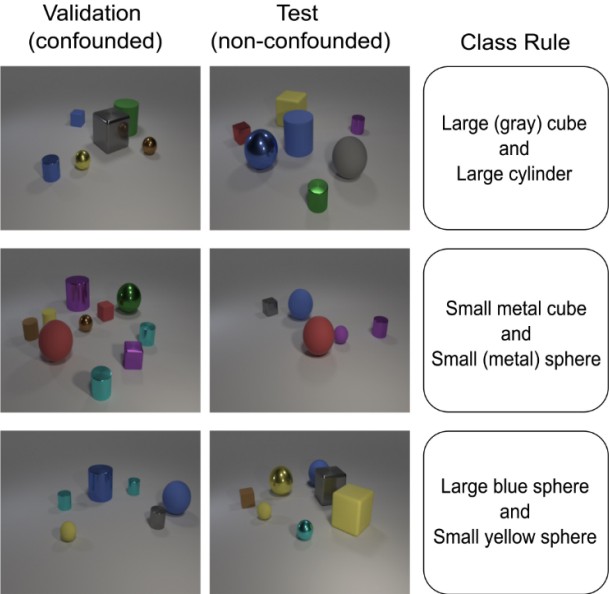

Figure 6: Figure from (Stammer et al., 2021).

## F Metrics

### F.1 Explanation consolidation.

**Ridge regression classification.** For evaluating the separability of learned explanations, we provide the accuracy of a ridge regression (RR) model that is fitted on a set of tuples consisting of explanations from a trained learner and the corresponding ground-truth (GT) class labels of the underlying image. The RR model is fitted on a training set and tested on an additional, held-out set of explanations (and corresponding class labels).

This evaluation acts as a proxy of the separability of learned explanations. The higher the RR accuracy on the test set the better the separability between explanations. For each learning configuration in our evaluations we train a RR model separately on the explanations from the five differently seeded models.

**Encoding analysis.** The Inter- vs Intraclass Explanation Similarity (IIES) is defined as:

$$\text{IICS} = \frac{1}{K} \sum_k^K \frac{\frac{1}{M} \sum_i^M d(z_i^k, \mu^k)}{\frac{1}{K} \sum_{j, j \neq k}^K d(\mu^j, \mu^k)}$$

Essentially, this metric estimates in how far the explanations stemming from samples of one class are close to another compared to the explanations of samples from other classes. The encoding of a pretrained model, $h$, provides the encoding space in which this similarity is assessed. The lower the values of IICS the better separable are the data for $h$.

Here $z_i^k$ corresponds to the encoding of the explanation of a sample $i$ from class $k$. This encoding is provided by an additional model, $h$, via $h(e_i) = z_i^k$, where $e_i$ is a provided explanation of sample $i$ from a learner $f$. $h$ is identical in architecture to the learner of which the explanations are being evaluated, however $h$ was separately pretrained only on the original task. Specifically, for evaluating explanations from the CNN configurations, $h$ corresponds to an identical CNN that was trained for image classification as in the vanilla configurations. For evaluating the NeSyCL configurations a NeSyCL was pretrained for classification as in the NeSyCL vanilla setting. In both cases $h$ was provided with a random seed different from those used in the original training setups.

Furthermore, $\mu^k$ corresponds to the average encoding over all samples of class $k$ (where for notations sake we assume $M$ samples in each class, although this can vary in practice). $d(x, y)$ represents a distance metric between $x$ and $y$, where we have used the euclidean distance in our evaluations. We divide the distance within one class by the average distance between the encoding mean of class $k$ and those of all other classes, corresponding to an estimate of the distance to all other class encodings. Finally this is averaged over all classes.

## F.2 Faithfulness

For comprehensiveness, parts of the input are removed that correspond to important features as identified by the explanation. As a result, the model should be less accurate in its predictions. In the case of sufficiency, one removes those input features which were deemed unimportant according to the explanation. Hereby, the model should not lose much accuracy. Notably, the original sufficiency and comprehensiveness metrics of (DeYoung et al., 2020) were introduced in the context of NLP in which input sequences are considered as discrete inputs. However, removing input features from continuous inputs such as images presents an issue (Hooker et al., 2019) as measured differences due to pixel removal may reflect the influence of the modified, out-of-distribution input rather than faithfulness of the explanation. For this case, we modified the metrics for the CNN configurations (*i.e.*, for explanations that are in a continuous form) to approximately compensate for this effect. For evaluating explanation faithfulness we thus provide results for CNN-LSX (and vanilla CNN) via the continuous adaptation of both metrics (denoted as $\text{COMP}_{cont.}$ and $\text{SUFF}_{cont.}$) and for NeSyCL-LSX (and NeSyCL vanilla) via the original comprehensiveness and sufficiency definitions (denoted as $\text{COMP}_{discr.}$ and $\text{SUFF}_{discr.}$). We formalize these in the following.

We follow the notation for $\text{COMP}_{discr.}$ and $\text{SUFF}_{discr.}$ of Chan et al. (2022). For this, $x$ denotes an input sample. We denote the predicted class of $x$ as $c(x)$, and the predicted probability corresponding to class $j$ as $p_j(x)$. Assuming an explanation is given, we denote denote the input containing only the $q\%$ important elements as $x_{:q\%}$. We denote the modified input sequence from which a token sub-sequence $x'$ are removed as $x \setminus x'$. Comprehensiveness and sufficiency for discrete explanations are finally defined as:

$$\text{COMP}_{\text{discr.}} = \frac{1}{|B|} \sum_{q \in B} \frac{1}{N} \sum_{j=1}^{N} (p_{c(x_j)}(x_j) - p_{c(x_j)}(x_j \setminus x_{j:q\%}))$$

$$\text{SUFF}_{\text{disc.}} = \frac{1}{|B|} \sum_{q \in B} \frac{1}{N} \sum_{j=1}^{N} (p_{c(x_j)}(x_j) - p_{c(x_j)}(x_{j:q\%})).$$

Where $N$ here represents the number of data samples in the evaluation set. In our evaluations we set $B = \{1, 5, 10, 20, 50\}$ as in the original work of DeYoung et al. (2020).

For computing comprehensiveness and sufficiency scores based on continuous explanations we first compute the comprehensiveness and sufficiency when a percentage $q$ of the top input elements (*e.g.*, pixels) are set to the median value of all input elements of the evaluation set. In comparison to the definition of $\text{COMP}_{discr.}$ and $\text{SUFF}_{discr.}$ of DeYoung et al. (2020) for the adaptation to continuous explanations we base the metrics on class accuracy rather than class probabilities. We denote these alternative computations as:

$$\hat{\text{COMP}}_{\text{cont.}} = \frac{1}{B} \sum_{q \in B} \text{acc}(f(X \setminus X_{:q\%}^{\text{median}}), y)$$

$$\hat{\text{SUFF}}_{\text{cont.}} = \frac{1}{B} \sum_{q \in B} \text{acc}(X_{:q\%}^{\text{median}}, y).$$

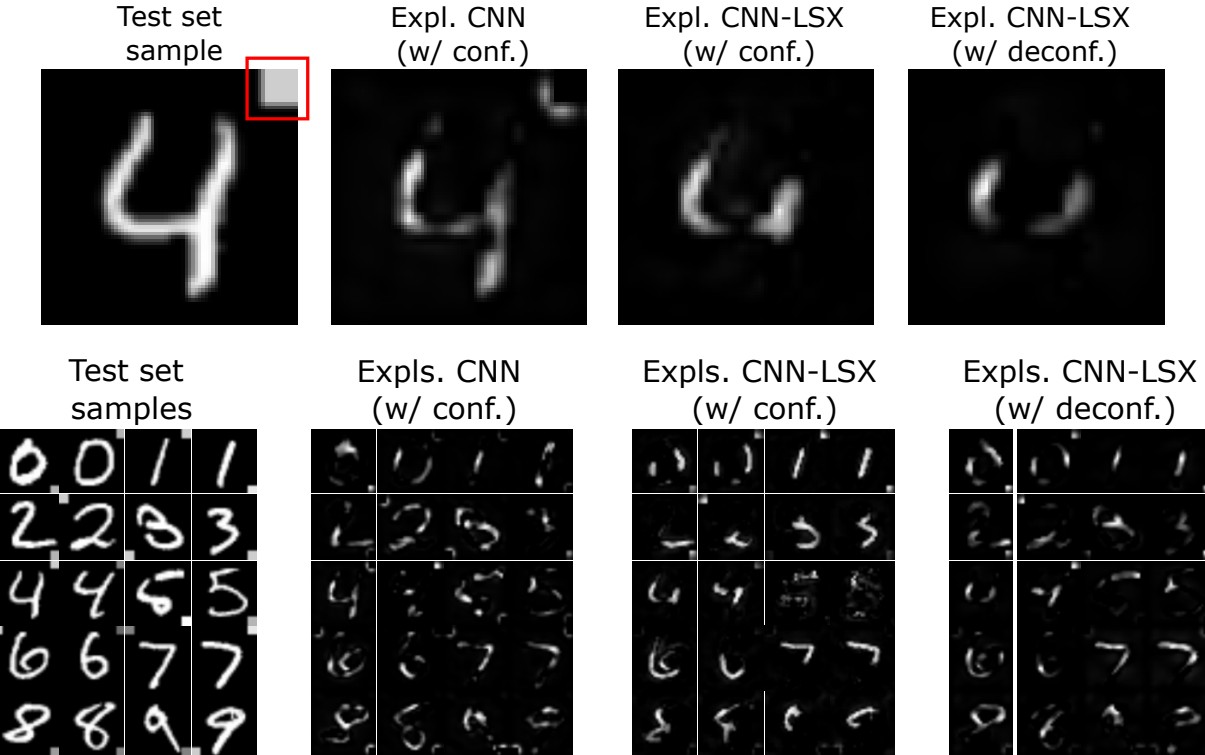

Figure 7: Example importance map (InputXGradient) explanations from the different CNN configurations on De-coyMNIST. The top row show an original test set sample and the corresponding explanations for CNN (w/ conf.), CNN-LSX (w/ conf.) and CNN-LSX (w/ deconf.). The bottom row shows the same setup for 20 randomly selected test set samples. In red we have highlighted the confounding factor of the specific example. Note that the CNN-LSX models (both trained w/ conf. and w/ deconf.) do not indicate importance of the confounder in their explanations.

Here, $\mathrm{acc}(f(X), y)$ corresponds to the accuracy score of a models prediction given input data, $f(X)$, compared to the ground truth labels, $y$. $X_{:q\%}$ corresponds to the full dataset in which everything but the top $q\%$ of each samples input elements were set to the median value of the dataset and $X \backslash X_{:q\%}^{\mathrm{median}}$ where the top $q\%$ of each samples input elements were set to the median value of the dataset.

Next we compute the same metrics, but when removing randomly chosen $q\%$ of the input elements by setting them to the median value. We denote these computations as $\hat{\mathrm{COMP}}_{cont.}^{rand}$ and $\hat{\mathrm{SUFF}}_{cont.}^{rand}$. Finally, we subtract these from the original values, leading to:

$$\mathrm{COMP}_{\mathrm{cont.}} = \hat{\mathrm{COMP}}_{cont.}^{rand} - \hat{\mathrm{COMP}}_{cont.}$$

and

$$\mathrm{SUFF}_{\mathrm{cont.}} = \hat{\mathrm{COMP}}_{cont.}^{rand} - \hat{\mathrm{COMP}}_{cont.}$$

### F.3 Self-unconfounding: Sample Explanations

Fig. 7 presents exemplary explanations from the different CNN configurations on the DecoyMNIST dataset. Specifically, we provide images from original test set samples (left), explanations of the baseline CNN (w/ conf.) (second to left), explanations of the CNN-LSX (w/ conf.) (second to right) and explanations of the

CNN-LSX (w/ deconf.) (right). The explanations correspond to the InputXGradient importance maps. The top row represents images for a single sample, where the red box in the test sample image indicates the confounder. We observe that both LSX configurations do not put any importance on this confounder for this sample. The bottom row shows the same setting for altogether 20 randomly selected test set images (two per class). Importantly, we observe a greatly reduced confounder importance in the explanations of the LSX configurations, though this is not fully removed (consistent with the accuracy results of Tab. 2).

Fig. 8 presents exemplary explanations ($e_{z_i}$) from the different NeSyCL configurations on class 1 images of the CLEVR-Hans3 dataset. Over four randomly chosen class 1 training images we observe that the baseline NeSyCL model puts great importance on the confounding factor of class 1 (*e.g.*, the gray color of large cubes, highlighted in red in the figure) the LSX based models both ignore this factor and even indicate the original groundtruth class rule (a large cube and large cylinder, highlighted in blue in the figure) despite never having received any explicit feedback on this. These qualitative results further indicate the confounding mitigation results observed in Tab. 2.

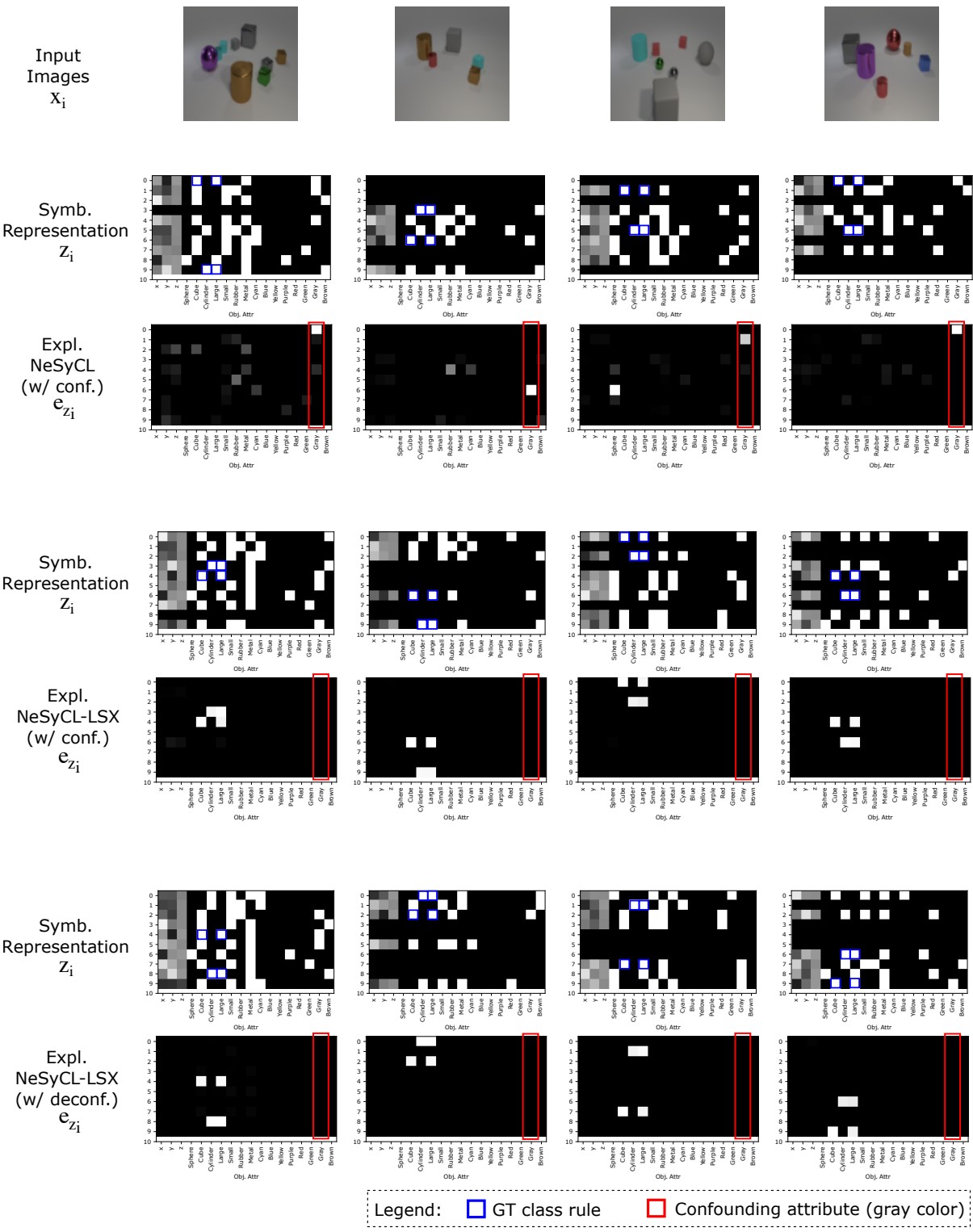

Figure 8: Example explanations from the different NeSyCL configurations for class 1 images of CLEVR-Hans3. Specifically, we provide images of the integrated gradients-based explanations, $e_{z_i}$. The first row depicts original images of four randomly selected training samples that belong to class 1. The second, fourth and sixth row depicts the symbolic representation, $z_i$, of these images, as processed by the slot-attention-based perception module, where row four and six merely represent row-wise permutations of $z_i$ in row two. Row three depicts explanations of baseline NeSyCL (w/ conf.). Row five depicts explanations from NeSyCL-LSX (w/ conf.) and the last row depicts explanations from NeSyCL-LSX (w/ deconf.). In red we highlight the confounding object attribute of class 1. In blue we highlight the underlying rule of class 1 based on each sample.

