# OpenReview forum: "Learning By Self-Explaining"
_TMLR — Rejected by TMLR_

### Review · Reviewer_ioXA · 2023-10-09

**Summary Of Contributions:**

The paper proposes a workflow of exploiting existing explanation methods to improve the performance of ML methods. The core idea is that a good explanation should be useful for target tasks. Specifically, the paper considers two case studies in classification tasks; it proposes to perform an auxiliary “Reflect” task (e.g., training another instance of the same classifier on heatmaps produced by the explanation method) and, as a way to evaluate the quality of the explanations, the auxiliary information (e.g., the loss of auxiliary classification) is added to the original classification loss to retrain the original classifier. Experiments show the benefits of the proposed workflow.

**Audience:**

Yes

**Broader Impact Concerns:**

Nothing to say.

**Claims And Evidence:**

No

**Requested Changes:**

### Critical for acceptance

*Claims and descriptions of contributions.*

As indicated in the weaknesses, a major re-writing is required to be transparent about what is really done in the work. Some additional comments are as below.

- I think most discussions about self-reflection/refinement/explaining in Abstract and Intro could be removed without hurting readers’ understanding because the proposed workflow is at best a very rough analogy to a human being explaining to her/himself how the tasks were performed.
- If the author(s) want to keep the current title, then I think a down-to-the-earth explanation of “Self-Explaining” is needed (maybe you can say “We call our approach ‘self-explaining’ because we evaluate the explanations by performing a similar task to the base learner”).
- Referring to “Reflect” as a “module” gives an impression that something is built regarding this. But, the case studies just use existing methods as the “Critic”.   Similar comments apply to other “modules”.

*Writing.* Some parts of the paper should be condensed, while other parts expanded. The clarity is also a major concern.

- In Sec 1&2, the basic workflow is explained repeatedly at least 4 times in Figure 1, Algo 1, and the paragraphs. I think a figure and several paragraphs are enough. In particular, Algo 1 is just a description of the workflow and is far from an algorithm; it can be removed, and the symbols could be merged into Figure 1.
- I have a concern about the separation of the last 2 steps of the current 4-step workflow. As to my understanding, in the “Revise” step, the “Critic” and the base learner should be trained *together*, so the “Reflect” and “Revise” steps are actually done *at the same time*. But the current Fig 1 and the 4-step separation look as if a computed and *fixed* “score” is fed into “Revise”, leaving the readers wondering how the score, a constant, could change the training.  Maybe the “Reflect” and “Revise” steps are conceptually separated, but this needs more explanation.
- The two case studies in Sec 3.1, which are the real work done in the paper, are explained in a lack of details and are unclear.
    - In Fig 2, why the score has the form with a partial derivative? And this is different from what is explained in the main text.
    - In CNN-LSX, you set an iteration budget (which *limits how much* training data is used in evaluating the explanations) but also input the *whole* training data X_f into the Critic as a final step. This seems to be conflicting engineering choices? And doesn’t this make an overfitting to the training data because the data were used twice? Appendix A.1 has a different eq for this final step, which one is correct?
    - I read the paragraph on NeSyCL-LSX at least 3 times, but still cannot understand it clearly. The Appendix seems to have more details and you could move them to the main text. Some specific questions: 1) does the representation form change when you make the “conjunctive combinations”? 2) when you “group the resulting candidate rules by image class”, does it mean per image per rule, and many rules for each class? 3) What are the learner’s explanations E_f? Doesn’t the learner only have per-image representation? \bar{E}_max and E_f seem to have different dimensionality??
    - Regarding the above point, introducing the many representations involved using mathematical symbols would certainly help. You could also consider expanding Figure 3 and adding the representations.

*Code*. Because this is engineering and experimental work and I have concerns about reproducibility, the code is required for acceptance (could be provided at the time of (possible) acceptance, or a deadline is highly desirable if the author(s) need to polish the project).

### Good for improvements

*Evidence for deconfounding.* It is good to see the improvements in classification accuracy. But, for the claim of “mitigating the influence of confounding factors”, this is not enough. It is possible that the workflow produces a model with better but still confounded explanations. To further judge this claim, we need 1) a detailed explanation of the dataset, particularly how the confounding is introduced and what is the unconfounded dataset (this is a *required change*); 2) It is desirable to examine the explanations generated from the final model and confirm they are really deconfounded (at least to some extents).

*Sec. 3.2 on causal inference*. The connection is weak because you did not do any “interventional and counterfactual experiments”. And I also cannot see what is the “idea of true rationale generation” in the approach.

What are the “balanced accuracy scores” mentioned in the experimental setup?

### Minor


“for NeSyCL-LSX for CUB-10” → “for NeSyCL-LSX on CUB-10”

**Strengths And Weaknesses:**

### Stongnesses

The workflow seems to be practical and to improve generalization, faithfulness, and more.

The possibility of removing the confounded correlations learned by the models is interesting to me.

### Weaknesses
*Over-claim and far-stretching story*.

As indicated in my summary, I think referring to the contribution as a “learning paradigm” or “learning framework” is an over-claim. What is considered is how to use existing explanation methods to improve existing learning methods. And the main contribution is the two case studies showing how to choose the auxiliary “Reflect” task and its output, and how to integrate the auxiliary information back into the original learning method. There is no paradigm that reframed our understanding of learning and explanation, and also not a framework that others can simply take and use.

Related to the above, I think saying the approach is “self-explaining” and “self-refining” is a far-stretching story. Implementing the proposed approach actually requires much engineering consideration and work by humans (and that is why I called it a “*work*flow”). Overall, we do not have a single learning machine that is “self-explaining” or “self-refining”, but other people (and the author(s) themselves) can follow this workflow and possibly work out how to improve other learning methods using other explanation methods.

*Writing* is a major weakness of the paper. See ****Requested Changes**** for details.

The *evidence for deconfounding and the connection to causality* is weak. See ****Requested Changes**** for details.

---

> ### Author Response · Authors · 2023-10-21
> **Initial Response**
>
> We first off wish to thank the reviewer for their time and are happy that the reviewer agrees with the value
> of our experimental results.
>
> R: “On the core idea of LSX, contributions of this work and frameworks.”
>
> Answer: The core idea of LSX is not quite “to use existing explanation methods to improve existing learn-
> ing methods”. Rather, it is to utilize explanations within a model’s learning process whereby the model
> self-evaluates the quality of these explanations without the need of external (human) feedback. The under-
> lying assumption of this self-evaluation is that an explanation is considered good if the explanation aids in
> performing the original task. Overall this indeed represents a paradigm shift in the way that learning and
> explaining should be performed. Furthermore, the instantiations, one of our contributions, showcase how to
> “plug & play” our proposed framework for developing novel instantiations. Hereby, an instantiation is more
> than just choosing the Reflect task, but developing the whole set of LSX components, from Fit to Revise.
>
> R: “paradigm vs workflow”
>
> Answer: We disagree that our work only poses a workflow and not a paradigm. Up to now no work exists
> on integrating explanations into the learning process via a self-evaluating loop. Nor has the importance of
> explanations within the learning phase been considered overall in machine learning (ML) outside of human-
> interactions. Thus our work represents a novel view on how explanations and learning jointly interact, as
> also both other reviewers, particularly knwa, agree. This reviewer (ioXA) defines a workflow as something
> that “requires much engineering consideration and work by humans” and claims this to be the case in our
> work. According to this definition every paradigm in machine learning is in fact a workflow. Specifically,
> there are many “engineering considerations” to be made within the paradigm of supervised learning (SL)
> (architecture, cost function, optimizer, data settings, etc), yet all such algorithms have a specific typology
> and pattern of learning steps in common which represent the SL “paradigm”. Our work, in the same way,
> introduces such a typology and pattern for how to develop LSX algorithms. We have mainly based our work
> on the following definition of a paradigm (https://www.britannica.com/dictionary/paradigm). Perhaps
> the reviewer could provide their definition to clarify this issue.
>
> R: “Remove discussions about self-reflection/refinement/explaining.”
>
> Answer: We believe the motivation of self-explaining from human cognitive psychology sets a valuable stage
> for our work as reviewer knwa also very much agrees on. We do not claim LSX to be a replica of human
> self-explaining, nor is this necessarily our goal. However, there is sufficient evidence from studies on human
> intelligence on the benefit of learning by self-explaining. In addition there is an increasing trend in evidence
> from AI research that self-refinement/self-evaluation has benefits. As always there is room for improvements,
> however we here provide the first work in the direction to combine these ideas, where indeed our work provides
> clear evidence towards the benefit of learning by self-explaining also in AI models. We therefore argue that
> understanding the human intelligence side of this motivation is beneficial, if not necessary, to grasp the full
> picture of our work.
>
> R: “Definition on self-explaining.”
>
> Answer: We agree this could be beneficial and have added one into the introduction.
>
> R: “Condense Sec 1 2”
>
> Answer: Indeed, we have reworked these sections to make them more concise (specifically we have moved
> the algorithm to the appendix), at the same time we have tried to keep the clarity and understandability
> which reviewers kg6n and knwa have highlighted.
>
> R: “Referring to ’Reflect’ as a ’module’ gives an impression that something is built regarding this. ...”
>
> Answer: What does it mean to “build something regarding this” and why should this play a role in naming
> a module? The goal of the module names in our work is to provide a clear and intuitive representation for
> each subtask, independent of how it is finally implemented. E.g., the goal of Fit is to have a model be fit
> to the training data. In the same way, the goal of Reflect is for the critic to reflect on the explanations, i.e.
> evaluate the quality of the explanations. Perhaps the reviewer could rephrase this remark?

---

> ### Author Response · Authors · 2023-10-21
> **Initial Response II**
>
> R: “Separation of Reflect and Revise steps.”
>
> Answer: In fact, they are not trained together. The learner is not optimised within Reflect, whereas the critic
> is not optimised during the Revise step. This is also the processing path in our instantiations. Thus we wish
> to distinctly separate the phase in which the critic creates a revisory feedback signal and a phase in which
> the learner incorporates this feedback signal. E.g. in CNN-LSX the learner incorporates the accumulated
> loss signal from the critic, where in NeSyCL-LSX the ranking score is used to enforce the explanation with
> the highest score. Thus, for the sake of incorporating different forms of feedback “score” acts as a general
> term here. However, we agree the text could be more clear on these steps of the framework, specifically
> when introducing the instantiations. In accordance with the feedback of reviewer kg6n on this topic we are
> working on providing this and will let the reviewers know once this is complete.
>
> R: “Unclear descriptions of instantiations, particularly NeSy-LSX.”
>
> Though reviewer kg6n positively remarked on the understandability of these descriptions, we are indeed
> working on updating both descriptions of the instantiations.
>
> R: “Partial derivative in Fig 2.”
>
> Answer: The partial derivative mainly acts to indicate that the score in CNN-LSX represents the backprop-
> agated classification loss of the critic, as indicated in Sec. 3.1.
>
> R: “On meaning of iteration budget, whole training set in final step, overfitting.”
>
> Answer: Actually, the iteration budget does not limit how much data is being used, but how many loops of
> Explain-Reflect-Revise are to be performed. During which, however, the learner and critic sets remain the
> same. We don’t find it conflicting, rather Xc is a subset of Xf at training time to reduce the computational
> cost of these iterations. However, for a final step we increase the number of samples in Xc for improved
> results. Thus in the final step the learner receives feedback twice based on the training set: once for class
> prediction and once for “good” explanations. This is similar to any other approach that jointly optimizes
> for several losses based on the same training set, e.g. variational autoencoders. One can even consider the
> explanatory feedback from the critic as a form of regularization. Indeed our results suggest the opposite of
> overfitting, i.e. better generalization performance despite haven seen the same data samples (and the same
> amount) as the baseline.
>
> R: “Equation in A.1”
>
> Answer: Both equations are correct. The equation of A.1 represents a variation for a very last revise step
> after the iteration budget has been reached for CNN-LSX. Up to this step the equation of 3.1 CNN-LSX is
> applied.
>
> R: “Code availability.”
>
> Answer: We fully agree on the importance of the reproducibility of science. For this reason we had added
> the code for NeSy-LSX as a supplemental zip file which the reviewer should have received. Moreover, the
> code will be publicly available after acceptance.
>
> R: “Evidence for deconfounding”
>
> Answer: First off, we do not claim that LSX models are fully independent of the confounders influence, but
> simply that LSX helps “mitigate” (i.e. reduce) the influence, which our evidence shows. In fact, Decoy-
> MNIST (cf. Ross et al. 2017 “Right for the Right Reason”) and CLEVR-Hans3 (cf. Stammer et al. 2020
> “Right for Right Nero-Symbolic Reasons”) were specifically designed such that the test set accuracy of a
> model gives a distinct indication of how strongly that model is influenced by the confounder. The goal
> of these datasets is thus to be able to provide strong quantitative results on the influence of confounders
> that don’t require the visual inspection of (in worst case cherry-picked) explanations. We have added some
> information on the datasets to the Appendix and will nevertheless add example model explanations soon
>
> R: “Sec. 3.2 on causal inference.”
>
> Answer: Thank you, we see now that this section actually reads more as “possible connections for future
> work” and distracts from the main points of our work. We have therefore moved this discussion to the section
> on future work.
>
> R: “balanced accuracy score”
> Answer: These are simply the classification accuracies of Tab. 1.

---

> ### Author Response · Authors · 2023-10-25
> **Detailed instantiation descriptions**
>
> Dear reviewer,
>
> We have now completed integrating the feedback on the descriptions of the instantiations:
>
> R: “Unclear descriptions of instantiations, particularly NeSy-LSX.”
>
> Answer: Thank you for this feedback! Indeed, we have now greatly updated these sections. In the context
> of this we have among other things added details on both instantiations, reiterated over the mathematical
> notation as well as updated both instantiation figures. We have also updated the CNN-LSX figure by
> removing the partial derivative and this way making it more consistent with the text, as the reviewer
> had remarked. We believe this overall has greatly improved the understandability of these contributions.
> However, please let us know if specific points remain unclear.
>
> R: “Separation of Reflect and Revise steps.”
>
> Answer: In the context of updating the instantiation sections we have now provided more details that
> illustrate the difference in these two learning steps. Let us know if it still remains unclear.

---

> ### Author Response · Authors · 2023-10-30
> **Added explanation examples for deconfounding results**
>
> We have updated the supplementary material with additional example explanation images for the two confounded datasets, DecoyMNIST and CLEVR-Hans3, further undermining our findings of Tab. 2. The reviewer can now find these images in App. D.3 (and referenced in the main text).
>
> Overall, the reviewers comments have lead to great improvements in terms of structure, writing and presentation. Please do let us know if any concerns remain. We are looking forward to the reviewer's response.

---

> > ### Comment · Reviewer_ioXA · 2023-11-15
> > **Thanks for the rebuttal**
> >
> > I have read the rebuttal and revised paper, but still have some concerns.
> > ### About the over-claim
> > I don't think arguing about the meaning of the words (e.g., "paradigm") is very helpful, but we could check how it is used in the ML community. Searching "paradigm of supervised learning" on Google Scholar, there are only 438 results (while there are 1,680,000 results for "supervised learning"), and this is strong evidence that the community does not usually refer to supervised learning as a "paradigm". Then how could the current approach qualify as a "paradigm"? Also, if the author(s) insists on the meaning of the word, your reference in fact agrees with me to some extent by saying "paradigm" can "be copied". This is what I meant by "others can simply take and use". Moreover,  the word "topology" you used to refer to the approach is in fact more like my "workflow" than a "paradigm", both "topology" and "workflow" will conjure up a flow chat, like Fig 1, in our mind.
> >
> > Setting aside the "talk" above, I suggest that if the author(s) can "code" up and document yet another case study before the possible acceptance, better not again on classification, then this concern will probably be addressed.
> >
> > ### Unclearness
> > The part on NeSyCL-LSX is better now, though I am not very sure due to the limited time to check. Some unclearness remains.
> >
> > If the Reflect and Revise are not optimized together, then my question remains that, "if a computed and fixed 'score' is fed into 'Revise'",  "how the score, a constant, could change the training"? On the other hand, you said, "the learner’s parameters are updated based on both classification losses: the one from the learner given the training images Xf and that from the critic given the explanations of Xc." This seems to agree that Reflect and Revise are optimized together, so I am really confused.
> >
> > I still do not understand the final step of Revise. I am very familiar with VAEs, and I think the statement is false that "This is similar to any other approach that jointly optimizes for several losses based on the same training set, e.g. variational autoencoders."

---

### Review · Reviewer_kg6n · 2023-10-16

**Summary Of Contributions:**

This paper introduces a novel learning framework - Learning by Self-Explaining (LSX), which enhances model performance through self-explanation. The authors conducted experiments on multiple tasks and datasets, showing that LSX can improve model performance. Overall, this is an interesting and valuable paper, but there are some areas that need improvement.

**Audience:**

Yes

**Broader Impact Concerns:**

I don't have any questions at the moment.

**Claims And Evidence:**

Yes

**Requested Changes:**

1.Insufficient Comparative Experiments: The authors only compared the performance of LSX with baseline algorithms on different tasks, and do not provide a comparative discussion about  Explainable ML  in the experimental and method sections. Including this would better illustrate the effectiveness of the approach.

2.It would be beneficial to include more examples that demonstrate the content of self-explanation and its corresponding scores. This would help in providing a clearer understanding of the learning process and how self-explanation affects the model's performance.

3.Need for Results on Cutting-Edge Datasets or Tasks: It is necessary to include experimental results on more cutting-edge datasets or tasks to support the efficacy of the LSX framework for complex problems.


Additional Issues:

●a. In the abstract, it is mentioned that "an explanation is considered 'good' if the critic can perform the same task given the respective explanation." However, it is unclear why the critic would perform the task if its primary role is evaluation.

●b. In the "Obtain explanations (Explain)" paragraph, using the same symbol 'f' to represent the explanation model as the learner may create ambiguity. Clarification is needed.

Overall, this paper presents a promising approach but needs more extensive experiments, comparative analysis with explainable ML approaches, and clarification of certain concepts to strengthen its contributions.

**Strengths And Weaknesses:**

1.Innovative Approach: LSX is an innovative learning framework that helps learners improve their performance by self-explaining their models and utilizing a critic for reflection. This paradigm of using an auxiliary model for reflection and feedback to enhance learning is novel.

2.Clear Explanations: The author's descriptions of the method architecture for both differentiable and non-differentiable feedback cases in the method section are clear and easy to understand.

3.Effective Experiments: The experimental results appear to be effective, and the self-explanations seem reasonable.

---

> ### Author Response · Authors · 2023-10-21
> **Initial Response**
>
> We thank the reviewer for their valuable time and are happy our work represents an “interesting and
> valuable paper”. We are further delighted that the reviewer agrees on the novelty of the paradigm, finds the
> descriptions of both the typology and instantiations clear and the experimental results effective.
>
> R: “Comparative analysis with explainable ML approaches.”
>
> Answer: Specifically, the XAI aspect in LSX is “only” one component of LSX and LSX per se is explainer-
> agnostic. Nor do XAI approaches perform any form of self-refinement. Thus, we are unsure of what and
> how to compare to XAI here. Could the reviewer try rephrasing this remark? However, we have added a
> more comparative discussion on XAI in the related works.
>
> R: “Include more examples that demonstrate the content of self-explanation and its corresponding scores.”
>
> Answer: If we understand this correctly, the reviewer is asking for more details and explicit examples of
> the reflection/evaluation process, how the score is computed and integrated into the learner. We are still
> working on this and will keep the reviewers posted.
>
> R: “Additional results on Cutting-Edge Datasets or Tasks”
>
> Answer: We fully agree that investigating the benefit and possible instantiations of LSX for other tasks than
> image classification is very important future work. However, we wish to highlight the complexity of the de-
> confounding task showcased by the baseline model performances in Tab. 2, but also the baseline performances
> in Tab. 1 that suggest that even the small data classification task is not as simple as assumed. Scaling our
> instantiations up to be able to handle different, e.g. larger, data sets to us represents an engineering aspect,
> e.g. replacing the learner models with a more suitable architecture. Such experiments would however neither
> prove nor disprove LSX’s principle eligibility, which is already provided by our experimental evidence on
> two substantially different instantiations. Nevertheless, we are currently investigating the possibilities for
> experiments on a third, novel LSX instantiation, which is based on multimodal transformers for VQA. We
> hope to be able to provide results soon and will keep all reviewers posted.
>
> R: “It is unclear why the critic would perform the task if its primary role is evaluation.”
>
> Answer: Correct, the critic’s main role is to evaluate the explanations of the learner. However, if we recall
> from our introduction “a ’useful’ explanation should provide important information for the task”. Thus we
> let the critic evaluate the explanations by performing the task given the explanations.
>
> R: “Same symbol ’f’ to represent the explanation model.”
>
> Answer: This is a misunderstanding. Rather, Explain() is the notation of the explanation function that
> receives the learner, f , and critic dataset, X_c, as arguments (in a similar way as for the Fit() function).

---

> ### Author Response · Authors · 2023-10-25
> **Details on self-explaining, score and revision**
>
> Dear reviewer,
>
> In the context of updating the sections on our instantiations we have now also put more focus on
> concrete examples of the self-explanation, score computation and score integration into the learner. The reviewer
> can now find these updates both in the corresponding text and figures. We feel this has made it much more
> understandable. Please let us know if you agree on this.

---

> ### Author Response · Authors · 2023-10-31
> **Novel results on VQA task**
>
> The reviewer can now find details on novel experiments on VQA-X representing both a cutting-edge dataset and task (visual-question answering) in App. B and C.3 in the updated version of the paper.

---

### Review · Reviewer_knwa · 2023-10-17

**Summary Of Contributions:**

The paper discusses the concept of "Learning by Self-Explaining (LSX)". The main idea behind LSX is that a learning module (referred to as the 'learner') performs a given task (e.g., image classification) and then provides explanations for its decisions. An internal "critic" module then assesses the quality of these explanations in relation to the original task. If the explanations are satisfactory, they can be used to refine the learning process. The authors have outlined four main components of LSX: Fit, Explain, Reflect, and Revise. The paper presents different implementations of LSX for two distinct learner models. The research finds that LSX can enhance the generalization abilities of AI models, reduce data requirements, and lead to more accurate and task-specific model explanations. In summary, there are 3 main contributions:
* Introduction of LSX: A new learning paradigm for machine learning that emphasizes self-refining by using explanations.
* Diverse Implementations: The paper showcases various instantiations of LSX, illustrating the flexibility of the sub-models and learning modules.
* Empirical Evidence: The research provides substantial experimental results on different datasets and metrics, highlighting the potential of LSX. They particularly note that LSX improves the generalization capabilities of base learning models, especially in scenarios with limited data. It also effectively addresses the influence of confounding factors, leading to clearer, task-specific model explanations.

**Audience:**

Yes

**Broader Impact Concerns:**

There is no ethical concern as far as I know.

**Claims And Evidence:**

No

**Requested Changes:**

1. It would be better to give some examples of the representation of the explanation in Section 2 Obtain explanations (Explain). For example, language, heatmap, etc.
2.	Information about the explanation in CNN-LSX is missing. The author says “The explain module is realized with the post-hoc, differentiable InputXGradient method”, but there are no details about the explanation. I notice that the details are provided in A.1, but I think it is important to be mentioned in the main context.
3.	In section 3.1, the authors say that “the model parameters are updated based on both classification losses”. I am not sure what “model” represents here. Does it mean the learner or both the learner and the critic?
4.	How is the explanation dataset X_c built for the results in Table 1 and Table 2? What is the ratio between X_c and X_f?
5.	In section 4.2 Self-unconfounding, the authors investigate the results of deconfounding. It would be better to briefly introduce what the spurious correlation looks like in the Decoy-MNIST and CLEVR-Hans datasets.
6.	In the de-confounding experiments, there is no explanation for the conclusion “This result suggests great practical implications, as it does not require prior knowledge on the confounding factors.” In the w/ conf. setting, if all training data contain spurious correlation, it is actually impossible for the model to identify the true causality. My understanding is that, if all training data contain spurious correlation, the so-called “spurious correlation” in this dataset is just an explanation from humans and the “shortcut” is the true correlation that the model should learn. Then a good model should fit well on that “spurious correlation”. Could the authors provide further intuition about the empirical improvement?

**Strengths And Weaknesses:**

**Strengths:**
1. The motivation of using human intelligence to benefit learning methods is convicting as there is a lot of evidence showing that self-debiasing and self-instructing could improve the performance of large language models.
2. The proposed framework is neat and easy to understand. Figure 1 helps me capture the structure and components of the proposed framework.
3. Using the self-explain framework for mitigating confounders is an interesting direction. As humans usually reason about the true causality with self-reflection, it is possible to improve the performance of deep learning models with a similar capability.

**Weaknesses:**
1. The conclusion of “generalization abilities of AI models” comes from only two models and four classification datasets (MNIST, ChestMNIST, CLEVR, CUB-10). What is the reason for selecting a CNN model and a neural-symbolic model? I think the title and the conclusion are too ambitious given the current evaluation protocol in this paper.
2. It seems that the authors propose a universal framework for improving the performance of models by using an additional critic. However, I feel that the design of the algorithm varies case by case and modules need to be carefully selected for different tasks and learners. With such case-by-case designs, the improvement over baselines may come from other aspects rather than self-explanation. For example, in the CNN-LSX instantiate, the explanation is basically an additional engineered feature (not verified if it is useful for all tasks) and the final training procedure just doubles the size of the model capacity and adds a loss to make the model learn from the new feature. If I only look at this design, it is hard for me to connect it with a self-explain framework. Therefore, I think that there is a gap between the proposed universal framework and specific instantiates.
3. The authors say “Note that for NeSyCL-LSX for CUB-10, we replaced the slot-attention perception module with a pre-trained Inception-v3 network” in section 4.1. Does this pre-trained model contain additional knowledge of other datasets? Is it still a fair comparison of the baseline?
4. The authors say that “In practice, we found that it was necessary to reinitialize the critic with each LSX iteration” in the appendix. It seems that it will take too much time if X_c is large.

---

> ### Author Response · Authors · 2023-10-21
> **Intitial Response**
>
> We wish to thank the reviewer for their time. We are happy the reviewer has identified the key aspects of
> our framework, they agree with the motivation from research on human intelligence and finds the framework
> “neat and easy to understand”.
>
> R: “Conclusion too ambitious; Universal framework vs case-by-case designs”
>
> Answer: In fact, we do not claim to be proposing a universal framework that provides benefits for all
> models. And we agree that providing experimental evidence for such a claim in general would be tricky, if
> not impossible. Rather, we are proposing a framework with a strong motivation of its benefits from human
> intelligence as well as the evidence from our own evaluations. Moreover, this novel paradigm is supported by
> parallels and evidence from similar branches of AI research from explanatory interactive learning [1, 2] (which
> however focuses on human explanatory revisory feedback) to recent self-reflective generative approaches [3]
> (which however do not yet explicitly focus on explanations within the reflective process). The evidence from
> our two instantiations indeed suggest the potential of LSX and, certainly, the goal of future research will be
> to consolidate these findings. Such accumulated evidence will be the best indication of LSX’s benefits for AI
> models, where we have laid the initial ground work with this paper.
>
> Furthermore, this issue addressed by the reviewer is a fundamental issue that holds also for other learning
> paradigms such as self-supervised learning (SSL). SSL was not proposed as a universal framework to solve all
> tasks of AI. However, the experimental evidence gathered thus far suggests benefits within a specific scope.
> Also here design choices can greatly influence performance improvements (e.g. over supervised learning) or
> absences thereof.
>
> We have carefully reformulated claims where necessary to reflect these statements and added a paragraph
> on limitations in the conclusion.
>
> R: “Choice of cnn and nesy”
>
> Answer: Our goal was to choose a diverse subset of current AI model approaches to showcase LSX. Both
> the CNN and NeSyCL are standard approaches within their subfields and particularly yield very different
> design opportunities for instantiating them in LSX. This way, we could show two very different instantiations,
> despite the same base task (image classification).
>
> R: “Pre-trained Inception-v3 network for CUB.”
>
> Answer: We apologize, in fact, for NeSyCL for CUB10 we only replaced the pretrained slot attention module
> with a pretrained Inception-v3 module, where “pretrained” here means that these modules where pretrained
> on the relevant data (CLEVR or CUB) for predicting the initial concepts (e.g. object shape and colors
> or bird wing color etc.). This is a standard approach for such NeSy/ concept-based models. The baseline
> models contained these pretrained components just as the LSX instantiations did, therefore leading to a fair
> comparison. We have updated the wording in the experimental setup section to make this more clear.
>
> R: “Computational cost of reinitializing the critic with each LSX iteration.”
>
> Answer: The computational load of this instantiation is far from perfect. However, our research has not
> yet focused on such things as efficiency and we consider this to be one of the goals of future work. Possible
> alternatives could e.g. be parameter efficient approaches such as retrieval based systems. In fact, we had
> performed this as a means to prevent overfitting on the side of the critic. Certainly other forms of overfitting
> prevention should be possible. We have reformulated this statement accordingly.
>
> R: “Examples of the representation of the explanation in Section 2 (Obtain).”
>
> Answer: We have added some remarks on this in the Obtain section.
>
> R: “Information about the explanation in CNN-LSX is missing”
>
> Answer: We have added this in the CNN-LSX section.
>
> R: “Clarifying: “The model parameters are updated based on both classification losses”.”
>
> Answer: Indeed we were referencing the learner here and have updated the sentence.
>
> R: “X_c in tables 1 & 2”
>
> Answer: We have added this information into the experimental setup section of the paper as well as the
> appendix.
>
> R: “Introduce the confounders for the readers.”
>
> Answer: Yes we agree, we have added information and figures into the appendix and main text.
>
> R: “Further intuition about the empirical improvement in de-confounding?”
>
> Answer: The reviewer can now find a detailed intuition on this in the paper within the deconfounding
> evalautions as well as some additional remarks in the conclusion.
>
> [1] Schramowski et al. (2020) “Making deep neural networks right for the right scientific reasons by interacting
> with their explanations”. Nature Machine Intelligence.
> [2] Brack et al. (2023) “ILLUME: rationalizing vision-language models through human interactions”. ICML.
> [3] Asai et al. (2023) “Self-RAG: Learning to Retrieve, Generate, and Critique through Self-Reflection”.
> CoRR.

---

### Author Response · Authors · 2023-11-17
**Response to AE's (and Reviewer's) feedback**

We thank the AE (and all reviewers) for their advice and are happy they agree on the potential of
our novel learning approach.

We have indeed followed both of the AE’s suggestions and have updated the paper in the following:

- Added two new pages of extra analysis and evaluations on potential failure cases of LSX to the Appendix
(App. C) and referenced this in the limitations section of the main text.
- Moved and greatly extended the limitations section in the main text from the Conclusion into a Discussions
section.
- Rewrote, shortened and moved the “general perspectives” (previously Sec.3.2) into the Discussions section
(i.e. after the evaluations).
- Performed rewriting throughout the paper, particularly focusing on the wording and presentation.

Overall, we feel this has immensely improved the clarity, foundation and structure of our work and we greatly
thank the AE and reviewers for their help and feedback. Please let us know if anything remains unclear
despite these efforts.

---

### Decision · Action_Editor_FWwA · 2023-11-19

**Recommendation:** Reject

**Comment:**

This paper considered the concept of Learning by Self-Explaining (LSX). The main idea behind LSX is that (1) a learning module (referred to as the *learner*) performs a given task (e.g., image classification) and then provides explanations as for its decisions. (2) An internal critic module then assesses the quality of these explanations in relation to the original task.Then explanations can be further used to refine the learning process, which can be viewed as a regularization.  The paper presents different implementations of LSX for two distinct learner models.

Most reviewers (including me) acknowledged the contributions and highlights within the paper. I appreciate the effort to explore a potential new direction by learning a better model through the explainability criteria. I think this could be a better way to construct a transparent model in AI.

However, after reading the rebuttal and revised paper, most reviewers still do not recommend acceptance at this time, despite the authors' tremendous efforts. Most importantly, many reviewers still feel that the **paper scope needs to be carefully narrowed down to avoid over-claiming**. I checked Sections 1-3 again and agreed with these feelings. The followings are some notable concerns:

- [Major] LSX can be viewed as a new concept/framework. However, if the main objective is to introduce such a novel concept, it should include certain in-depth analysis to illustrate the insights of the proposed concept. For example, in the GANs (Generative Adversarial Networks) paper, the learning framework is formerly formulated as a min-max game (or adversarial process). An alternative way is to provide extensive empirical results in diverse datasets and tasks (e.g., *A Generalist Agent*,or *Mixup* paper) to justify the practical utility.

- [Minor] About the notation. The paper has several inconsistent notations on page 3 (problem statement). For example, does X include label y? Is there any validation set? If I understand correctly, there should exist four datasets: training data for the FIT part (X_f), validation data (X_c) for the critic part, validation data (in general) for hyperparameters, and hold-out test data? The audience might be confused by the different notations.

- [Minor] This paper claimed additional experimental benefits such as deconfounding, learning from few data. I agree that they could be potentially beneficial, while the reviewers found additional concerns rather than appreciations from the results. I think this is partially addressed by providing failure examples, but the paper needs revisions.

**Suggestions** I would suggest a major revision in the following possible ways.

1.  The title and abstract can be narrowed down, such as Learning by Self-Explaining in Image Classification. Then abstract and paper could have significant concentration without introducing too many general concepts.
2.  If the author wants to keep the title LSX, I would suggest that this paper could follow the idea of the mixup paper, which is a very good example to illustrate how to demonstrate the practical utility of a new concept.
3.  It is great to consider both neural and neural symbolic instantiations. However, the writing part should be given additional attention. For example, you can decouple them in Section 3.1.
4. Some parts should be put in the appendix. A possible reason for over-claiming lies in the introduction of general but not strongly relevant concepts (such as system 1/2, self-refinement). Reviewers may feel a bit underwhelmed by the gap between the actual contribution and such general concepts.

Based on the reviewers' comments and my readings, we find it a bit difficult to accept this paper under a minor revision. I strongly recommend that you incorporate some feedback to carefully rewrite the paper. I do not think you need significant additional experiments, but rather a reorganization and rewriting.

**Audience:**

Yes.

**Claims And Evidence:**

No. Please refer to the comments for details.

**Resubmission Of Major Revision:**

The authors may consider submitting a major revision at a later time.